# Discovery of surrogate agonists for visceral fat Treg cells that modulate metabolic indices in vivo

Ricardo A Fernandes[1†], Chaoran Li[2†§], Gang Wang[2], Xinbo Yang[1], Christina S Savvides[1], Caleb R Glassman[1], Shen Dong[1], Eric Luxenberg[3], Leah V Sibener[1], Michael E Birnbaum[1], Christophe Benoist[2], Diane Mathis[2‡*], K Christopher Garcia[1,4‡*]

[1]Departments of Molecular and Cellular Physiology and Structural Biology, Stanford University School of Medicine, Stanford, United States; [2]Department of Immunology, Harvard Medical School; and Evergrande Center for Immunologic Diseases, Harvard Medical School and Brigham and Women's Hospital, Boston, United States; [3]Department of Electrical Engineering, Stanford University School of Engineering, Stanford, United States; [4]Howard Hughes Medical Institute, Stanford University School of Medicine, Stanford, United States

*For correspondence:
diane_mathis@hms.harvard.edu
(DM);
kcgarcia@stanford.edu (KCG)

†These authors also contributed equally to this work
‡These authors also contributed equally to this work

Present address: §Department of Microbiology and Immunology, Emory University School of Medicine, Atlanta, United States

Competing interests: The authors declare that no competing interests exist.

**Abstract** T regulatory (Treg) cells play vital roles in modulating immunity and tissue homeostasis. Their actions depend on TCR recognition of peptide-MHC molecules; yet the degree of peptide specificity of Treg-cell function, and whether Treg ligands can be used to manipulate Treg cell biology are unknown. Here, we developed an $A^b$-peptide library that enabled unbiased screening of peptides recognized by a bona fide murine Treg cell clone isolated from the visceral adipose tissue (VAT), and identified surrogate agonist peptides, with differing affinities and signaling potencies. The VAT-Treg cells expanded in vivo by one of the surrogate agonists preserved the typical VAT-Treg transcriptional programs. Immunization with this surrogate, especially when coupled with blockade of TNFα signaling, expanded VAT-Treg cells, resulting in protection from inflammation and improved metabolic indices, including promotion of insulin sensitivity. These studies suggest that antigen-specific targeting of VAT-localized Treg cells could eventually be a strategy for improving metabolic disease.

## Introduction

Foxp3$^+$CD4$^+$ T regulatory (Treg) cells restrain most types of immune response, guarding against run-away reactions (*Sakaguchi et al., 2008*). For many years, almost all Treg-cell studies focused on those circulating through or residing in lymphoid organs, so that our view of Treg-cell phenotype and function was heavily colored by the biology of this constellation of cells. More recently, some of our attention has turned to distinct populations of Treg cells that operate in non-lymphoid organs, where they not only rein in local immune responses, but also help maintain tissue homeostasis (*Panduro et al., 2016*). 'Tissue-Tregs,' as they have been termed, have distinct transcriptomes, T-cell receptor (TCR) repertoires, and dependencies that optimally arm them to survive and function within a particular tissue.

A paradigmatic tissue-Treg population is that found in visceral adipose tissue (VAT), in particular in the male gonadal fat depot (*Mathis, 2013*). According to a number of assays, VAT-Treg cells control local and systemic inflammatory and metabolic indices, promoting insulin sensitivity through influences on both inflammatory and parenchymal cells, notably macrophages and adipocytes. Like other tissue-Treg populations, VAT-Treg cells have a distinct transcriptome, driven primarily by the

'master-regulator' of adipocyte differentiation, PPARγ (*Cipolletta et al., 2012*). They also display a distinct, clonally expanded TCR repertoire that exhibits signs of clonal selection (*Feuerer et al., 2009*; *Kolodin et al., 2015*). The critical importance of the TCR for the accumulation and function of VAT-Treg cells was recently established using the vTreg53 TCR-transgenic (tg) mouse line, which has a TCR repertoire highly skewed for the specificity displayed by an expanded VAT-Treg clone (vTreg53) (*Li et al., 2018*). However, beyond its lack of dependence on the lipid-presenting CD1d molecule (*Kolodin et al., 2015*), little is known about the antigen(s) recognized by VAT-Treg cells.

More generally, the question of the antigen specificity of Treg cells remains open because very few peptide antigens that selectively activate them have been identified. How peptide-specific is Treg-cell function, and can Treg-cell ligands be used to manipulate Treg biology? The technical difficulties of identifying Treg-cell antigens are many, including the low precursor frequency of naturally occurring Treg cells, and experimental means of identifying peptide antigens in general. De novo identification of cognate peptides for TCRs of interest is still notoriously challenging and apart from a few examples for testis-cancer antigens and self-peptides recognized by neonatal thymic Treg cells, the identity of Treg TCR ligands remains elusive (*Leonard et al., 2017*; *Stadinski et al., 2019*; *Wang et al., 2004*). This reduced set of Treg-specific antigens largely restricts the phenotypic characterization and manipulation of activated Treg cells to broad, polyspecific, stimulation regimes. Furthermore, while polyspecific stimulation can produce Treg-mediated suppressive activity, antigen-specific activation appears to elicit stronger responses in autoimmunity or graft rejection treatment (*DuPage and Bluestone, 2016*; *Tang et al., 2004*; *Yamazaki et al., 2006*).

Here we sought to gain insight into the antigen specificity of Treg cells in a well-characterized system of VAT-localized Treg cells with known functional metrics that can be queried using candidate surrogate ligands. We developed an $A^b$ yeast peptide library and screened for recognition by the TCR displayed by the VAT-Treg clone vTreg53, identifying a range of surrogate peptides (SPs) that robustly stimulated the original clone both in vitro and in vivo. Exploiting one of these SPs, we devised a 'vaccination' protocol that was able to ameliorate metabolic indices. These studies collectively reinforce the antigen-specific nature of Treg function and highlight the continued importance of ligand discovery to explore Treg function and devise antigen-specific Treg therapeutic strategies.

## Results

### Evolution of the peptide-$A^b$ yeast-display library

In order to discover peptides recognized by the vTreg53 TCR, we performed an unbiased, affinity-based screening of a peptide-MHC yeast library. For this approach, we first had to engineer a large peptide-$A^b$ library in yeast cells. Based on previous work on the peptide-$E^k$ and other pMHC yeast libraries (*Birnbaum et al., 2014*; *Brophy et al., 2003*; *Shusta et al., 1999*), we designed a single-chain construct composed of a 13-mer peptide, p3K, fused to the β1α1 domains and a c-Myc tag, connected by suitable Gly-Ser linkers, in-frame with the Aga2 protein for expression at the yeast surface (*Figure 1A*). While we were able to observe expression of this construct at the yeast surface, we could not detect binding to the cognate YAe-62.8 TCR (YAe)(*Dai et al., 2008*), suggesting that the peptide-$A^b$ construct was misfolded (*Figure 1—figure supplement 1A*). To overcome this issue, we generated a yeast library expressing a small number of randomly inserted mutations in the mini peptide-$A^b$ construct using an error-prone polymerase chain reaction strategy (PCR; up to 3 to 5 mutations per kbp), which we then selected with the YAe TCR.

Sequencing of the isolated clones after 4 rounds of selection using TCR-coupled streptavidin beads revealed two single-point mutations in the β-chain of all sequenced clones: Leu67Val and His46Tyr (*Figure 1A* and *Figure 1—figure supplement 1A,B*). Neither mutation appeared to impact the positioning of the peptide in the groove or interacted with the peptide anchor residues (*Zhu et al., 2003*). Furthermore, based on crystal structures for TCR interactions with cognate peptide-$A^b$ molecules, the two mutant residues were found to be outside of the peptide-$A^b$/TCR interface (*Figure 1A*; *Dai et al., 2008*; *Liu et al., 2002*; *Yin et al., 2011*).

To test if the newly evolved $A^b$ display platform could be used to deorphanize TCRs of interest, we generated a random-peptide library by incorporating the degenerate codon NNK at all but the anchor positions of a 13-mer peptide (corresponding to p1, p4, p6 and p9; *Figure 1A*), which we then selected using two TCRs of known specificity: YAe and 2W (*Figure 1B–D* and *Figure 1—figure*

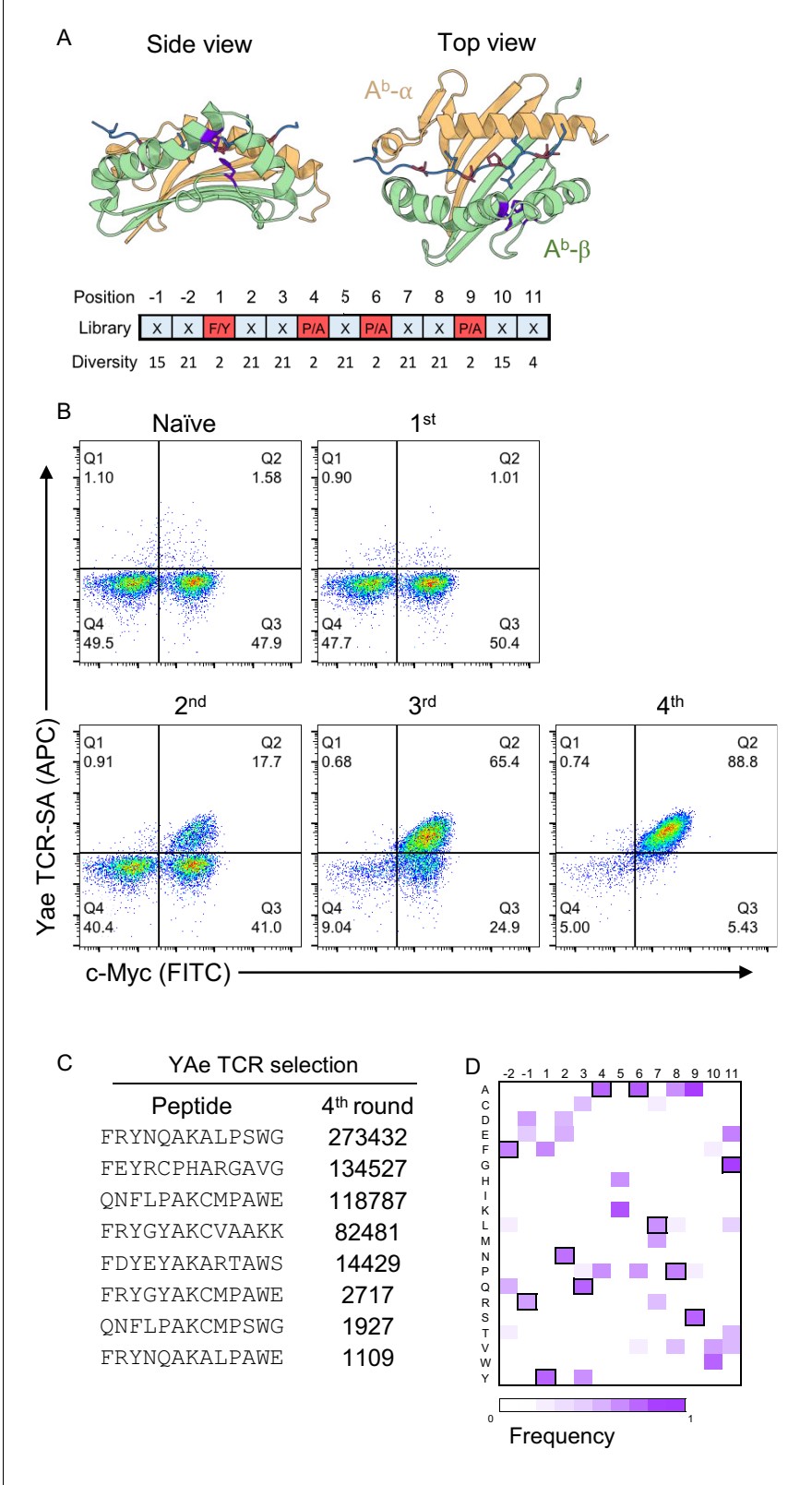

**Figure 1.** Development of a peptide-A$^b$ yeast library. (**A**) A degenerate codon with maximum diversity at TCR-facing positions (blue) and optimal residues at anchor positions (red) was used to generate a peptide-MHC yeast library containing up to $1 \times 10^9$ peptides. (**B**) To validate the library a proxy TCR of known specificity, YAe, was used to perform four consecutive rounds of selection. (**C**) Enriched yeast clones were isolated at all steps and

*Figure 1 continued on next page*

*Figure 1 continued*

deep-sequenced following appropriate plasmid purification and PCR amplification in order to quantify the peptide frequency. The top eight enriched peptides found in the fourth round of YAe selection are shown. (D) Positional frequency matrix based on the deep sequence of the fourth round of selection.

The online version of this article includes the following figure supplement(s) for figure 1:

**Figure supplement 1.** Evolution of peptide-Ab yeast display.
**Figure supplement 2.** Validation of the peptide-Ab yeast display platform.

*supplement 2A*). In each round of TCR-selection, the fraction of c-Myc and TCR-tetramer (coupled to streptavidin, TCR-SA) positive cells increased, suggesting the enrichment of a particular group of peptides (*Figure 1B* and *Figure 1—figure supplement 2A*). Deep sequencing of the DNA region encoding the peptide sequence from the isolated yeast clones for each of the four consecutive rounds of selection revealed a strong enrichment for a group of closely related peptides (*Figure 1C, D*). For example, nearly all peptides contained a Lys in the central, TCR-facing, p5 position, a residue that is critical for recognition by the YAe TCR (*Figure 1C,D*; *Dai et al., 2008*; *Huseby et al., 2005*). To confirm that the observed TCR-SA binding was strictly peptide-dependent, we substituted the central Lys residue at p5 by a Leu residue. This single-point mutation eliminated TCR-SA staining (*Figure 1—figure supplement 2B*). In other TCR-facing positions, such as p3, p7 or p8, we noticed a striking similarity to the peptides enriched from an insect peptide-A[b] library, namely Arg or Leu in p2 and p7, followed by Ala or Thr in p8 (*Crawford et al., 2004*). We followed a similar strategy with a second proxy TCR, 2W, and again found an increase in the fraction of TCR-tetramer positive cells in the third and fourth round of selection (*Figure 1—figure supplement 2A*). Furthermore, the 2W TCR showed a positive tetramer staining of the mini-A[b] platform displaying the cognate, p3K, peptide (*Figure 1—figure supplement 2C*). The dominant peptide in the 2W selection also showed positive 2W TCR-SA staining, which was lost after a single mutation of His to Leu (*Figure 1—figure supplement 2C*), confirming that binding is driven by peptide recognition. Taken together, the selection with the Yae and 2W TCR indicates that the newly generated peptide-A[b] library can be used to identify SPs for TCRs of interest.

## Identification of robust agonist surrogate peptides for the vTreg53 TCR

Next, we screened the peptide-A[b] library with the vTreg53 TCR, which was previously identified in a population of Treg cells present in epididymal VAT tissue (*Kolodin et al., 2015*; *Li et al., 2018*). The fraction of c-Myc-tag positive cells increased from ~29% in the naïve library to ~57% by the 4[th] round of the vTreg53 TCR selection (*Figure 2A* and *Figure 2—figure supplement 1*). However, staining with the vTreg53 TCR-SA tetramer showed a modest positive population (*Figure 2A*). Despite this result, deep sequencing of the four rounds of selection revealed strong enrichment of a group of highly similar peptides (*Figure 2B–D* and *Supplementary File 1*). Around 90% of all peptides found in the 4[th] round of selection had the KGPHAVQ/A sequence from p2 to p8. Lys at p2, and Val at p7 were present in all peptides, while the p5 position was dominated by the closely related His or Arg residues (*Figure 2B–D*). Furthermore, positive vTreg53 TCR-SA staining was detected for a single clone of peptide-A[b] displaying the most enriched peptide: LMFKGPHAVQAVG (*Figure 2E*).

Having found peptides recognized by the vTreg53 TCR using an affinity-based screening strategy, we next sought to identify robust agonist peptides using a T cell activation screen. For this approach, we tested the T cell activation potency of around 100 single-point mutant peptides for each of the two peptide sequences identified in the yeast-selection screen: LMFKGPHAVQAVG and TMYKNPRPVAATG, Fat7 and Fat15, respectively. (*Figure 3A,B*). Fat7 (yellow in *Figure 3A,B*) and Fat15 (magenta in *Figure 3B*) were both able to up-regulate CD69 expression following stimulation of Jurkat T cells transduced with the vTreg53 TCR in culture with peptide-pulsed K562-A[b] cells (*Figure 3—figure supplement 1A,B*). For both peptide libraries, the majority of single-point mutants eliminated activation or had a negligible effect when compared with the original Fat7 or Fat15 peptides (*Figure 3A,B*). However, a few single-point mutations based on Fat15, particularly those present in the p7 position, led to a marked increase in CD69 up-regulation (*Figure 3B*). Mutation of Val to Met or Trp at p7 induced the highest levels of CD69 expression (*Figure 3B,C*). The substitution of Pro to Leu in p4, an anchor position, also resulted in an increase in CD69 expression. Titration of

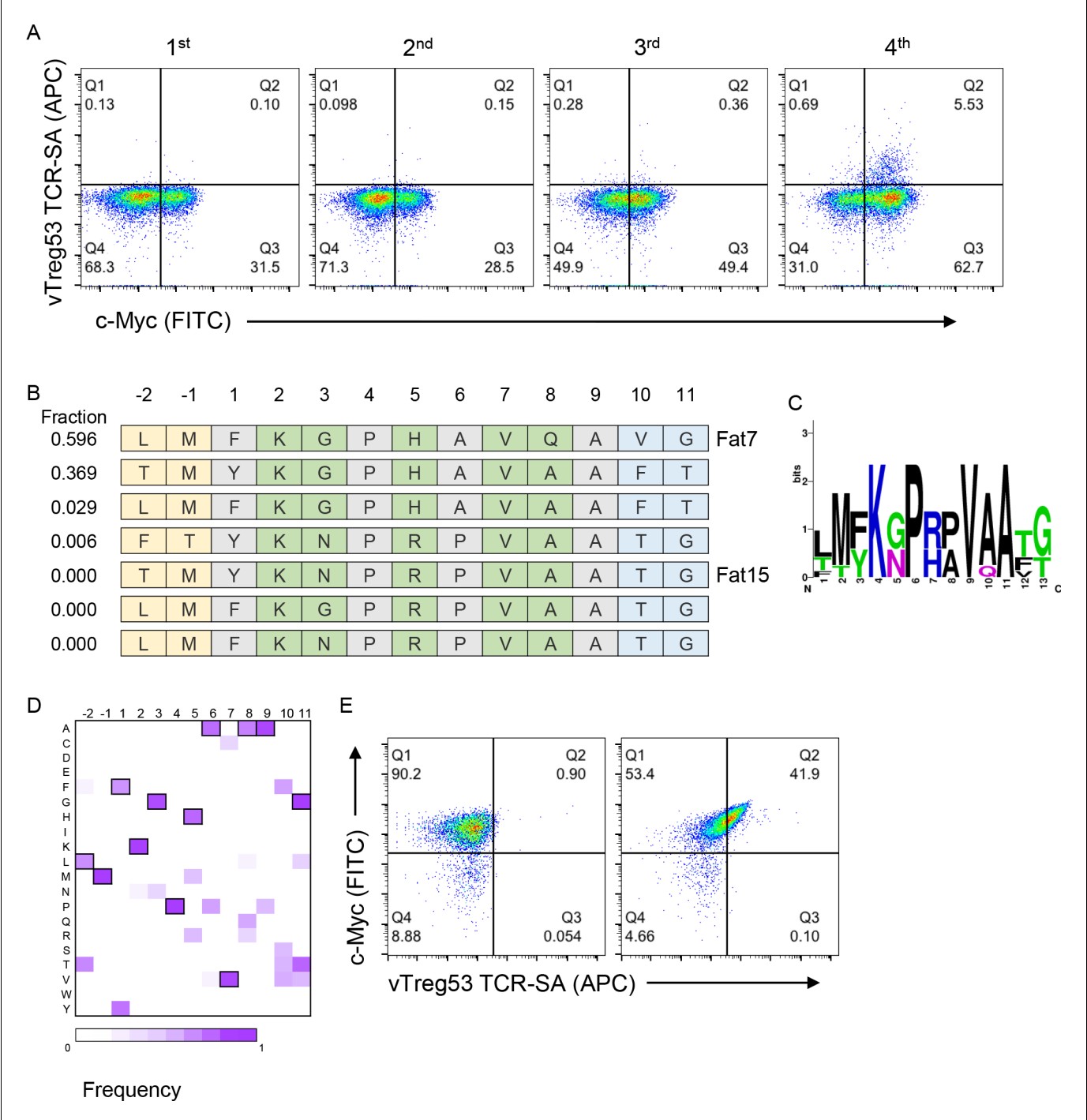

**Figure 2.** Identification of surrogate peptides recognized by the vTreg53 TCR. (**A**) The vTreg53 TCR was used to screen and isolate the peptide-MHC yeast display. vTreg53 TCR tetramer staining (500 nM of TCR-SA) together with c-Myc staining is shown for the naïve library and four consecutive rounds of selection. (**B**) Deep sequencing of the fourth round of selection reveals a small subset of peptides which share key residues at TCR facing positions (shown in green). (**C**) Positional frequency representation from the deep sequencing of the third round of selection with vTreg53. (**D**) Positional frequency matrix based on the deep sequencing of the fourth round of the peptide-MHC yeast selection. (**E**) A single peptide-MHC yeast clone, displaying the top enriched peptide, LMFKGPHAVQAVG (Fat 7; *right panel; left panel* shows staining for Eα control peptide), shows a positive Fat-TCR tetramer staining (500 nM final tetramer concentration). Data shown are representative of at least two independent experiments.

The online version of this article includes the following figure supplement(s) for figure 2:

**Figure supplement 1.** Evolution of peptide-Ab yeast display.

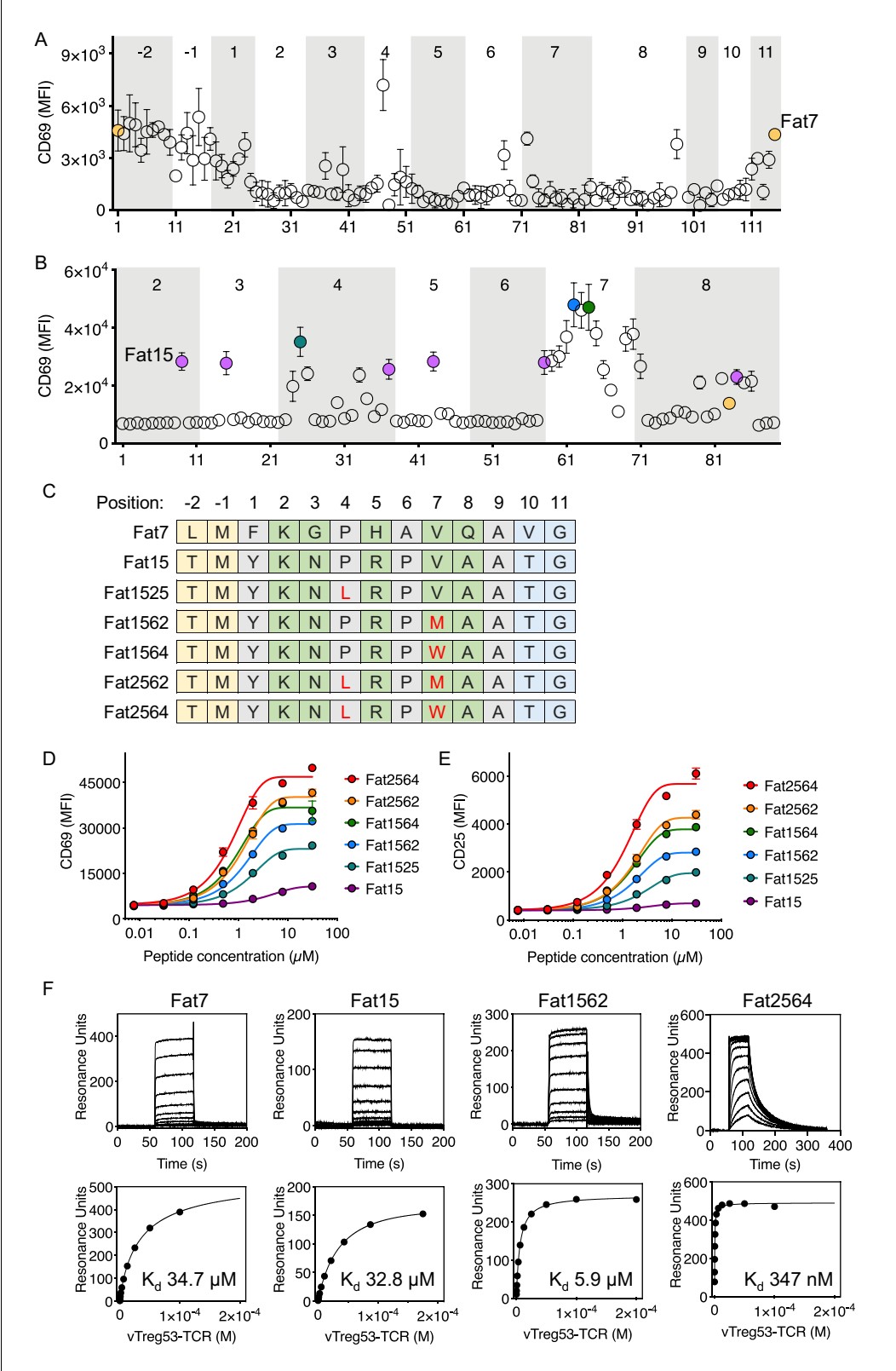

**Figure 3.** Identification of robust vTreg53 TCR agonists. Single-point mutants based on the Fat7 (**A**) or Fat15 (**B**) peptides were used to stimulate Fat-TCR transduced Jurkat T cells. Data shown are mean fluorescence intensity for CD69 up-regulation following overnight stimulation with the indicated peptides at 100 μM. (**C**) Potent SPs based on Fat15. Single-residue mutations from the Fat15 are shown in red. Anchor positions are shown in grey and potential TCR-facing positions are shown in green. A peptide titration of single- and double-point mutants from the Fat15 SP show up to ~4- (D; CD69)

*Figure 3 continued on next page*

*Figure 3 continued*

or ~8 fold (E; CD25) increase in $E_{max}$. (F) Surface plasmon resonance for Fat7, Fat15, Fat1562 and Fat2564 was used to determine the pMHC/Fat TCR affinity. All biotinylated pMHC were purified by SEC and immobilized in a streptavidin-coated SPR sensor chip from 200 up to 600 RU. Data are mean ± SD from $n$ = 2 (A, D, E) or 3 (B) biological replicates from 1 representative of 3 independent experiments. Data shown in (F) is representative of 2 independent experiments.

The online version of this article includes the following figure supplement(s) for figure 3:

**Figure supplement 1.** In vitro peptide stimulation of the vTreg53 TCR cells.

Fat15 and the peptides with Met at p7 (Fat1562) and Leu at p4 and Trp at p7 (Fat2564) revealed a decrease in $EC_{50}$ from 40.7 µM for Fat15 to 14.3 µM for Fat1562 and 8.9 µM for Fat2564 (*Figure 3D*). The CD69 $E_{max}$ was also increased by ~3 fold for Fat1562 and almost 5-fold for Fat2564 when compared with Fat15 (*Figure 3D*). The increase in activation for Fat1562 and Fat2564 was further confirmed by CD25 up-regulation, where these peptides induced a 4.1- and a 8.7-fold increase in $E_{max}$, respectively, over Fat15 (*Figure 3E*). A similar trend was found for the fraction of CD69+ and CD25+ cells (*Figure 3—figure supplement 1C*). The functional screen based on a mini-peptide library built around the Fat7 and Fat15 peptides allowed us to identify a group of closely related peptides, differing by 1 or two point mutations that showed substantive gains in T cell activation potency.

To characterize the binding kinetics and affinity ($K_d$) of key isolated peptides, we recombinantly expressed peptide-$A^b$ fusions for Fat7, Fat15, Fat1562 and Fat2564, and measured $k_{on}$, $k_{off}$ and $K_d$ using surface plasmon resonance. While all peptides appeared to show fast on-rates, typical for TCR/pMHC interactions, Fat1562 and Fat2562 exhibited the slowest $k_{off}$ and highest affinity ($K_d$ ~5.9 µM and 347 nM, respectively; *Figure 3F*). Fat7 and Fat15 showed both fast $k_{on}$ and $k_{off}$ and a $K_d$ of ~34.7 µM and ~32.8 µM, respectively (*Figure 3F*). Furthermore, the Fat15-$A^b$ and Fat1562-$A^b$ tetramers readily stained vTreg53-TCR-transduced Jurkat T cells, with Fat1562-$A^b$ showing a marked increase in the fraction of tetramer-positive cells over Fat15-$A^b$, in good agreement with the binding-affinity measurements (*Figure 3—figure supplement 1D*). The combination of an unbiased affinity-based selection from a very large pool of diverse peptides with a focused functional screen allowed us to identify robust surrogate agonist peptides that produced a strong T cell activation in vitro.

## SPs, especially Fat1562, induced robust proliferation and activation of vTreg53 TCR-tg Treg cells in vitro and in vivo

We next examined whether the SPs could also activate vTreg53 Treg cells with varying potency. To this end, we first labeled Treg cells from pooled lymph nodes of 6–8 week-old vTreg53 TCR-tg mice with CellTrace Violet dye and stimulated them with syngeneic antigen presenting cells (APCs) loaded with increasing concentrations of various SPs in the presence of IL-2 for three days. At a concentration of 1 µM, all SPs induced robust proliferation and activation of TCR-tg Treg cells, as measured by dilution of the CellTrace Violet dye (*Figure 4A and B*) and upregulation of CD44 (*Figure 4C*), respectively. Fat1562 induced the strongest down-regulation of CD62L, while Fat7 dampened CD62L levels only moderately at this high concentration (*Figure 4D*). In agreement with this finding, at a low dose of 0.01 µM, Fat7 was unable to promote any further proliferation and up-regulation of CD44 expression compared with the control group (no peptide; *Figure 4A–C*). Again, at this concentration, Fat1562 induced the strongest proliferation and down-regulation of CD62L in TCR-tg Treg cells (*Figure 4D*). Interestingly, Fat2564, the strongest SP for the vTreg53 TCR in the Jurkat system, had a relatively weak potency, which was comparable with that of Fat7 for primary TCR-tg Treg cells (*Figure 4A–D*), indicating that the optimal threshold for peptide stimulation was different for different T cell subtypes, or between primary cells and cell lines.

Since Fat1562 was consistently the strongest SP for vTreg53 Treg cells in vitro, we next investigated whether it could also promote proliferation of TCR-tg Treg cells in vivo. To this end, we intravenously (*i.v.*) transferred CellTrace Violet dye-labeled TCR-tg Treg cells into CD45.1+ congenically marked C57BL/6 (B6) mice, and on the next day immunized recipient mice subcutaneously (*s.c.*) with Fat1562 emulsified in complete Freund's adjuvant (CFA/Fat1562). Injection of CFA alone led to very little proliferation of transferred Treg cells in the spleen three days after transfer. However, when the

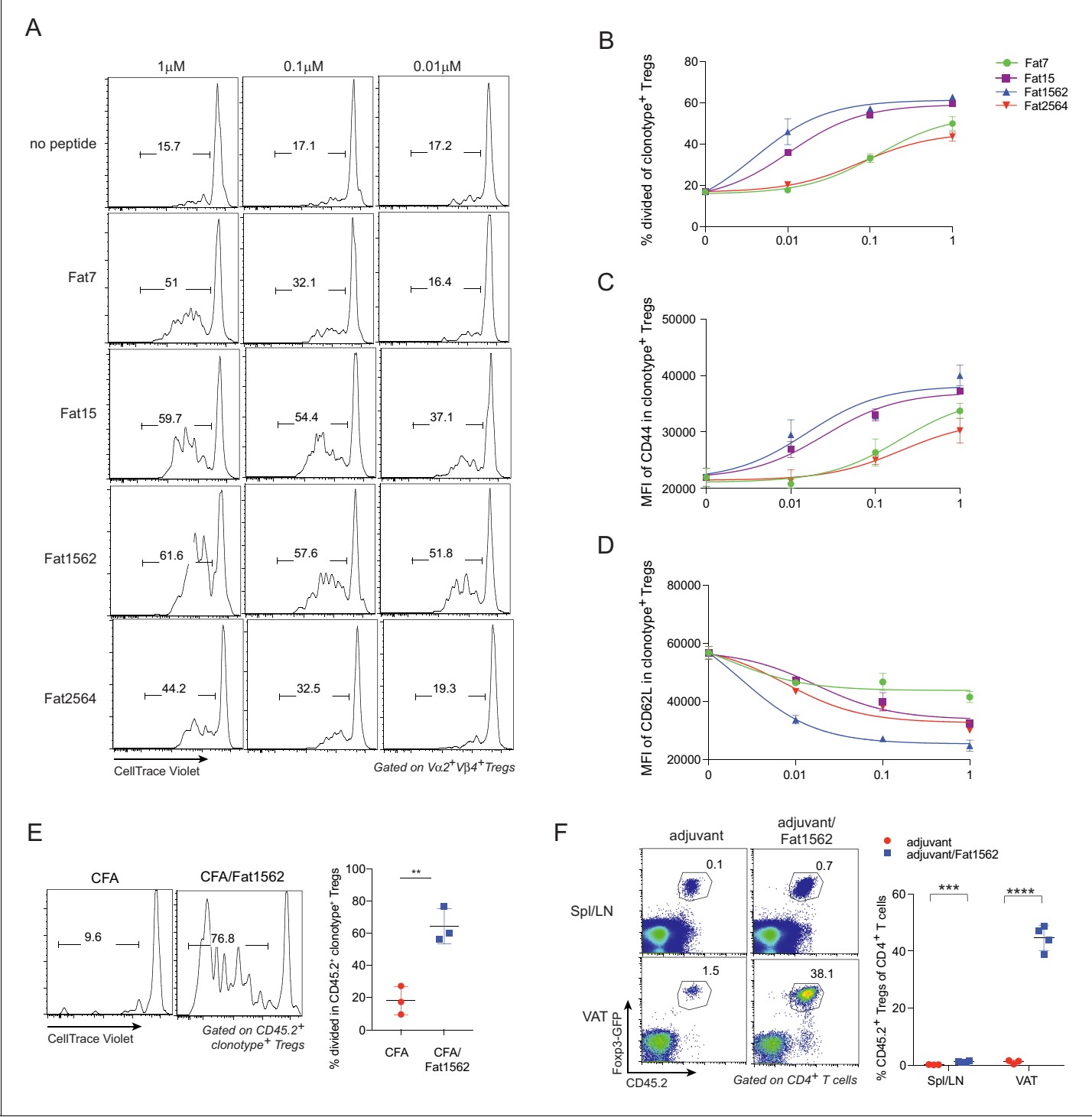

**Figure 4.** SPs induce proliferation and activation of vTreg53 Treg cells in vitro and in vivo. (A–D) Proliferation and activation of vTreg53 Treg cells by different concentrations of SPs for 3 days in vitro (n = 3). (A) Representative flow cytometric plot of cell division. (B) Summary of cell proliferation. (C) Mean flourescence intensity (MFI) of CD44 staining in clonotype[+] Treg cells. (D) MFI of CD62L staining in clonotype[+] Treg cells. (E) Proliferation of transferred CD45.2[+] vTreg53 Treg cells in the spleen of CD45.1[+] B6 recipient mice immunized *s.c.* with CFA or CFA/Fat1562 at day 3 (n = 3). (F) Expansion of transferred CD45.2[+] vTreg53 Treg cells in the Spl/LNs or VAT of CD45.1[+] B6 recipient mice that were primed with CFA/fat1562 *s.c.* and boosted with IFA/Fat1562 *i.p.* (n ≥ 3). Data are mean ± SD. Data shown are representative of at least two independent experiments.
The online version of this article includes the following figure supplement(s) for figure 4:

**Figure supplement 1.** Single cell TCR sequencing (scTCR-seq) analyses of fat1562-reactive endogenous Treg cells following immunization.

recipient mice were immunized with CFA/Fat1562, transferred $V\alpha2^+V\beta4^+$ (clonotype$^+$) Treg cells underwent robust proliferation (*Figure 4E*).

We next investigated whether Fat1562 could also expand TCR-tg Treg cells in VAT. We have previously reported that following transfer into B6 mice, vTreg53 TCR-tg Treg cells preferentially accumulated specifically in the VAT of recipient mice over the course of several weeks. Therefore, we transferred TCR-tg Treg cells into CD45.1$^+$ B6 mice, waited six weeks for the transferred cells to establish in VAT, and then immunized the recipients *s.c.* with CFA/Fat1562, followed by a boost with incomplete Freund's adjuvant (IFA)/Fat1562 intraperitoneally (*i.p.*) one week later. As expected, donor-derived TCR-tg Treg cells preferentially accumulated in VAT compared with spleen and lymph nodes (Spl/LN) of recipient mice injected with adjuvant alone (*Figure 4F*). Immunization with Fat1562 induced a 6- to 7-fold expansion of TCR-tg Treg cells in the Spl/LN, compared with adjuvant alone. Remarkably, it induced an even more substantial expansion (>25 fold) of clonotype$^+$ Treg cells in the VAT (*Figure 4F*). Therefore, the SPs identified through the yeast screen, especially Fat1562, were able to induce robust proliferation and activation of vTreg53 Treg cells in vitro and in vivo.

## Transcriptional analyses of vTreg53 TCR-tg Treg cells upon antigen stimulation

The identification of SPs for the vTreg53 TCR also allowed us to investigate what transcriptional programs were modulated when a Treg cell encountered cognate antigen in vivo. To this end, we performed RNA-seq analysis on donor-derived clonotype$^+$ TCR-tg Treg cells sorted from pooled Spl/LN or VAT of recipient mice following Treg transfer and Fat1562 immunization as described above. We first asked whether the clonotype$^+$ Treg cells expanded in VAT by Fat1562 were bona fide VAT Treg cells. As expected, compared with those in the lymphoid organs, clonotype$^+$ VAT Treg cells from mice immunized with adjuvant alone showed enriched expression of the previously defined VAT-Treg up-signature, while the down-signature was under-represented (*Figure 5A*, *left*). In mice immunized with adjuvant plus Fat1562, clonotype$^+$ VAT Treg cells still exhibited preferential expression of VAT-Treg signature genes (*Figure 5A*, *right*), indicating that the VAT Treg cells expanded by Fat1562 preserved the typical VAT-Treg characteristics.

We next performed Gene-Set Enrichment Analysis (GSEA) on the RNA-seq data to determine what pathways were modulated in VAT vs lymphoid-organ Treg cells upon antigen stimulation. In agreement with the remarkable expansion of clonotype$^+$ Treg cells in VAT by Fat1562 immunization, Fat1562-stimulated VAT-Treg cells showed significant enrichment in pathways and transcripts related to cell division (E2F targets, G2M checkpoint, MYC targets, Mitotic spindle) and glycolysis (*Figure 5B&D*, *left*). Several Treg signature genes such as *Lag3* and *Tigit* were also upregulated in Fat1562-stimulated VAT-Treg cells (*Figure 5D*, left). On the other hand, in lymphoid organs, antigen stimulation led to increased expression of genes associated with responses to inflammatory cytokines (IFN$\alpha$, IFN$\gamma$, and TNF$\alpha$), likely reflecting changes in the microenvironment of lymphoid organs with peptide immunization. But most interestingly, we observed an enrichment of genes involved in cholesterol homeostasis (*e.g.*, *Pparg*, *Nfil3*, and *Srebf2*), a hallmark of VAT-Treg cells (*Figure 5B&D*, *right*). *Pparg* mRNA levels were significantly increased in Spl/LN Treg cells when stimulated by Fat1562 in vivo but remained substantially lower than in Fat1562-stimulated VAT Treg cells (*Figure 5C*). We have previously identified that the spleen hosts a small population of PPAR$\gamma^{lo}$ Tregs that are precursor cells for VAT Tregs. These results fit with a model in which TCR stimulation upregulates PPAR$\gamma$ expression and/or promotes the induction of VAT-Treg precursor cells in the spleen. However, other stimulus, like IL-4, might also up-regulates PPAR$\gamma$ expression. These different possibilities are under further investigation. Collectively, these results indicate that Fat1562 was effective at expanding VAT Treg cells.

## A clonal response of endogenous treg cells to Fat1562 immunization in wild-type mice

We next investigated whether Fat1562 immunization of wild-type B6 mice would elicit expansion of particular Treg clones in VAT, and whether their TCRs would resemble the vTreg53 TCR. 16–20 week-old B6 mice were immunized with Fat1562 using the prime and boost scheme (*Figure 4—figure supplement 1A*), and the Fat1562-reactive CD4$^+$ T cells were characterized by staining with the Fat1562/A$^b$-PE tetramer. Interestingly, out of the 10 mice immunized, three showed detectable

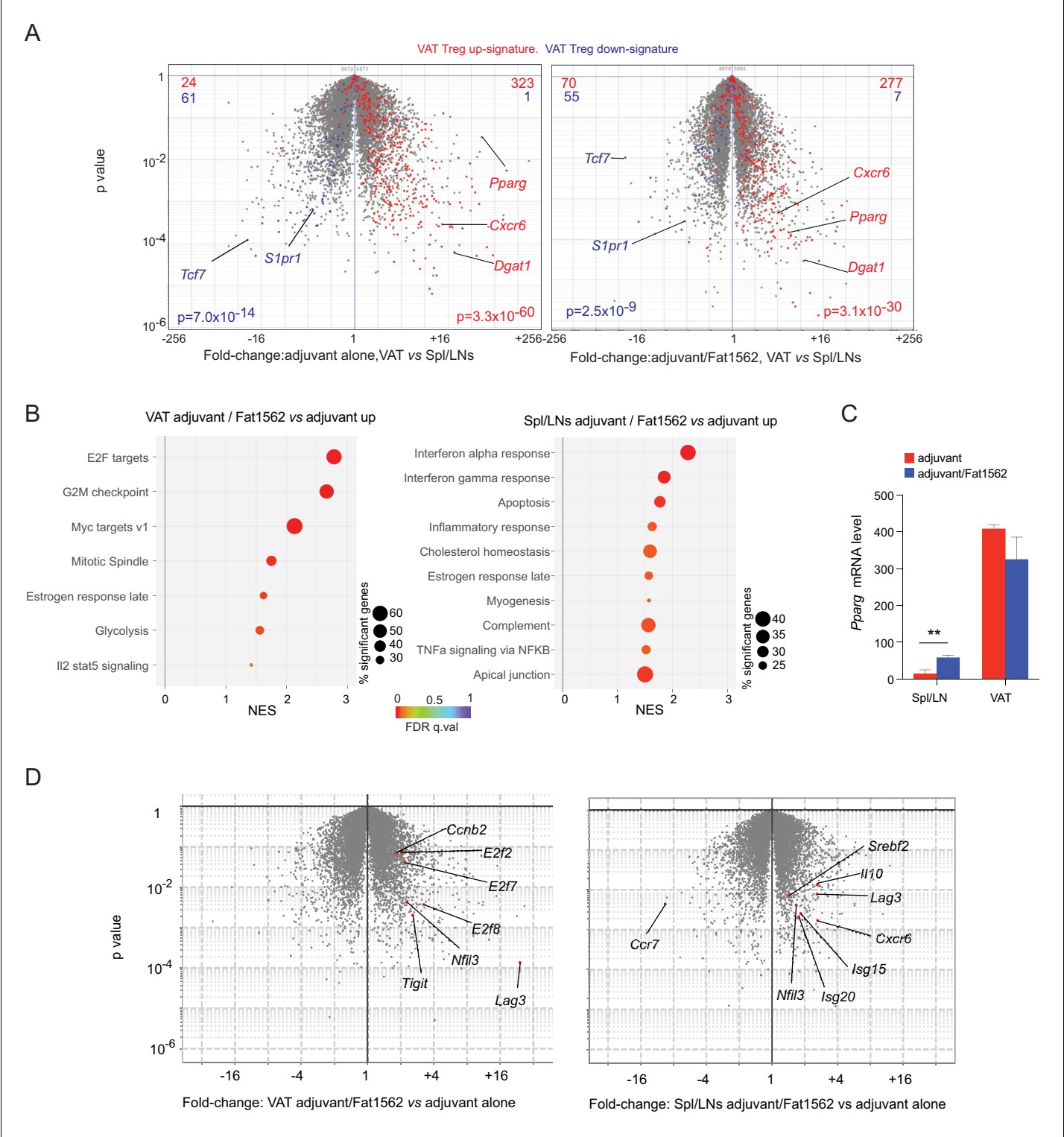

**Figure 5.** Transcriptional analyses of vTreg53 Treg cells stimulated by Fat1562 in vivo. (**A–C**) CD45.2⁺ vTreg53 Treg cells were transferred *i.v.* into CD45.1⁺ B6 mice. 6 weeks later, the recipient mice were immunized with CFA alone or CFA/Fat1562 *s.c.*, and one week later, boosted with IFA alone or IFA/Fat1562 *i.p.* CD45.2⁺ clonotype⁺ Treg cells were sorted from Spl/LNs or VAT of the recipient mice one week later for RNA-Seq. (**A**) Volcano plot comparing gene expression of transferred clonotype⁺ Treg cells in the VAT and Spl/LNs of recipient mice immunized with adjuvant alone (left) or with adjuvant/Fat1562 (right). VAT-Treg signature genes are highlighted in red (induced) or blue (repressed). The number of genes from each signature preferentially expressed by one or the other population are shown at the top. (**B**) GSEA analysis of top KEGG pathways (p<0.05) that are enriched in cells activated by adjuvant/Fat1562 compared with adjuvant alone in VAT (left) or Spl/LN (right). NES, normalized enrichment score. FDR, false

*Figure 5 continued on next page*

Figure 5 continued

discovery rate. (C) Normalized reads of *Pparg* transcript in CD45.2[+] clonotype[+] Treg cells sorted from mice immunized with adjuvant alone or adjuvant/ Fat1562. Data are mean ± SD. (D) Volcano plot comparing gene expression of transferred clonotype[+] Treg cells in mice immunized with adjuvant/ Fat1562 and adjuvant alone in the VAT (left) and spleen (right). Representative transcripts are highlighted.

expansion of the Fat1562/A[b] tetramer[+] Treg population in the Spl/LN and VAT, characteristic of the high variability of endogenous Treg cells' responses to specific peptide immunization (*Figure 4—figure supplement 1B&C*). We then sorted individual Fat1562/A[b] tetramer[+] Treg cells from the spleen and VAT of these 3 mice into 96-well plates, and determined the sequences of their TCR-β and TCR-α complementarity-determining region (CDR)3 s. Surprisingly, in each of the three mice, the Fat1562/A[b] tetramer[+] Treg population was almost entirely monoclonal (*Figure 4—figure supplement 1D*). Within each mouse, the Spl/LNs and VAT Treg clones had the same CDR3α and CDR3β. However, across different mice, none of the clones was shared, and their CDR3 sequences did not show any obvious similarity. This result is in agreement with our previous reports that the VAT-Treg clones naturally expanded are different in each mouse (*Kolodin et al., 2015*). Nevertheless, one of the Fat1562/A[b]-reactive clones had a CDR3α and a CDR3β highly reminiscent of the vTreg53 CDR3s, with the exact same CDR3α sequence and only one amino acid difference in CDR3β (*Figure 4—figure supplement 1D*). Therefore, although with some degree of variability, Fat1562 was able to expand an endogenous Treg population that highly resembled the original vTreg53 VAT-Treg clone.

## Fat1562 potently expanded transferred vTreg53 Treg cells in VAT, ameliorated local inflammation, and improved metabolic indices in mice fed a high-fat diet (HFD)

Chronic, low grade inflammation in VAT is one of the major drivers of obesity-induced insulin resistance, and VAT Treg cells are important for keeping this inflammation in check. We next asked whether we could use Fat1562 to expand VAT-Treg cells and suppress VAT inflammation during obesity. Given the high variability of Fat1562 to expand endogenous VAT-Treg cells, we decided to use Fat1562 to expand transferred vTreg53 TCR-tg Treg cells in recipient mice that were fed a HFD, and to determine whether this accumulation could consequently dampen obesity-associated VAT inflammation and metabolic dysregulation. To this end, we transferred vTreg53 TCR-tg Treg cells into CD45.1[+] B6 mice that had been fed a HFD for two weeks, immunized the recipient mice with CFA/Fat1562 *s.c.*, and *i.p.*-boosted the response with IFA/Fat1562 one week later. We then analyzed the composition of the immunocyte compartments in the spleen and VAT of recipient mice after another week of HFD feeding (*Figure 6A* and *Figure 6—figure supplement 1*). As expected, immunization with Fat1562 led to a 5- to 10-fold expansion of the transferred TCR-Tg Treg cells in the spleen, particularly for clonotype[+] Treg cells (*Figure 6B–D*). However, donor-derived cells still contributed less than 2% of the total Treg pool in the spleen of recipient mice. As a result, the total number of splenic Treg cells in recipient mice did not show a significant increase with immunization (*Figure 6F*). As expected, with adjuvant alone, transferred TCR-tg Treg cells preferentially accumulated in VAT compared with the spleen, but still made up less than 5% of the total VAT-Treg compartment in recipient mice (*Figure 6B*). In contrast, immunization with adjuvant plus Fat1562 led to substantial expansion of clonotype[+] TCR-tg Treg cells, reaching almost half of the entire VAT-Treg pool in recipient mice (*Figure 6B,C and E*). As a consequence, the total number of VAT Treg cells was also significantly elevated (*Figure 6G*).

In lean mice, VAT is enriched for anti-inflammatory immunocyte subsets, while obesity promotes differentiation/activation of pro-inflammatory immunocytes, particularly CD11c[hi] macrophages. Consistent with an important function of VAT-Treg cells in maintaining the balance between different types of immunocytes in the local microenvironment, immunization with Fat1562 significantly increased the proportion of eosinophils while reducing the frequency of CD11c[hi] pro-inflammatory macrophages (*Figure 6H*; gating strategy is shown in *Figure 6—figure supplement 1*). In agreement with dampened VAT inflammation, and although body weights were similar (*Figure 7A*), Fat1562 immunization of HFD-fed B6 mice that received vTreg53 Treg cells improved insulin sensitivity (*Figure 7B and C*, *Figure 7—figure supplement 1*), promoted glucose tolerance (*Figure 7D and E*), and reduced fasting plasma insulin levels (*Figure 7F*).

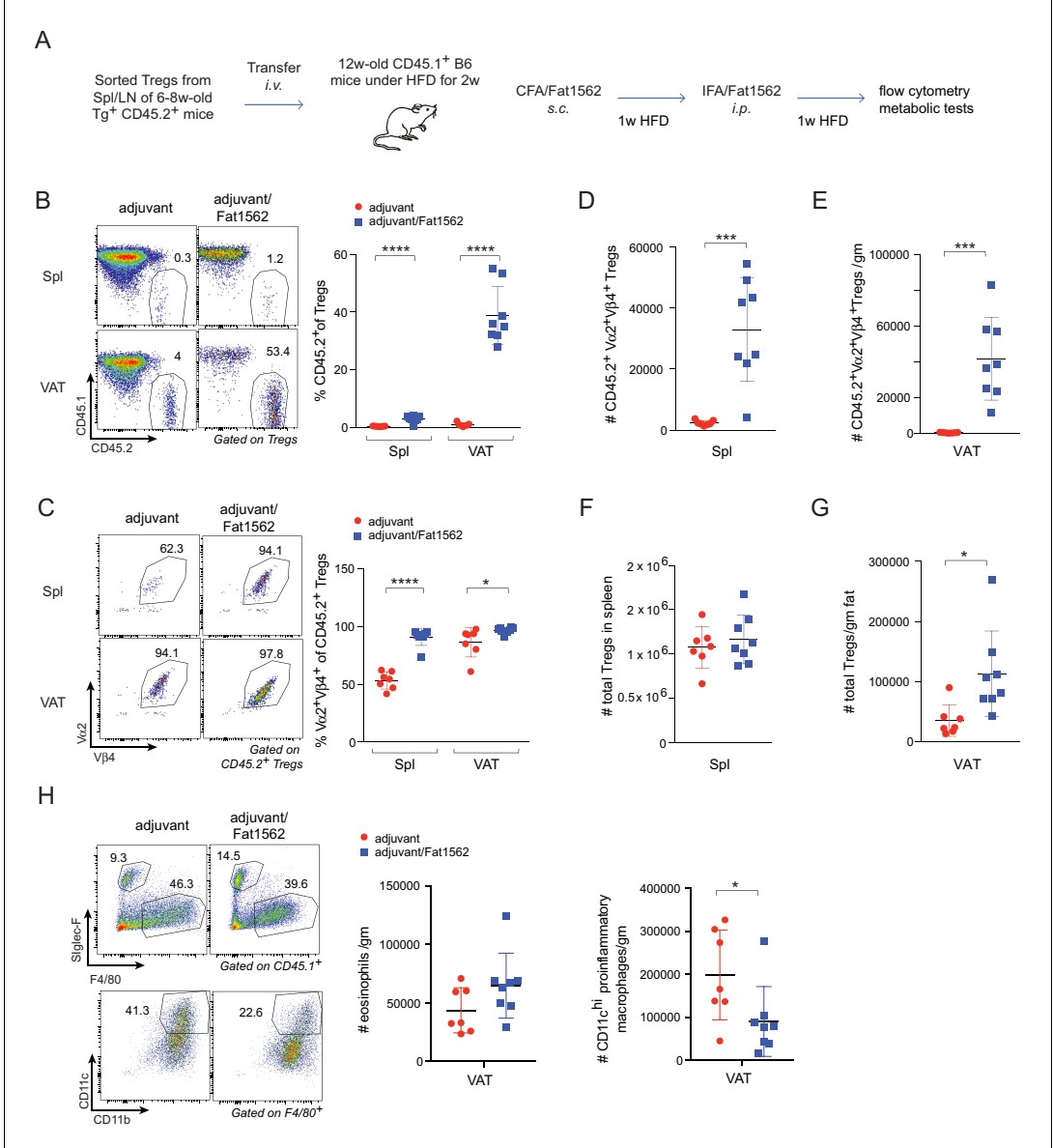

**Figure 6.** Immunization with Fat1562 expands transferred vTreg53 Treg cells and suppresses HFD-induced VAT inflammation. (**A**) Scheme of the transfer and immunization protocol. (**B**) Frequencies of CD45.2+ cells in Treg cells (n ≥ 7). (**C**) Frequencies of clonotype+ cells in transferred CD45.2+ Treg cells (n ≥ 7). (**D**) Number of clonotype+ CD45.2+ Treg cells in the spleen (n ≥ 7). (**E**) Number of clonotype+ CD45.2+ Treg cells in the VAT (n ≥ 7). (**F**) Number of total Treg cells in the spleen (n ≥ 7). (**G**) Number of total Treg cells in VAT (n ≥ 7). (**H**) Numbers of eosinophils and CD11chi inflammatory macrophages in VAT (n ≥ 7). Data are mean ± SD. Data shown are representative of at least two independent experiments.

The online version of this article includes the following figure supplement(s) for figure 6:

**Figure supplement 1.** Gating strategy.

We previously reported that VAT Tregs progressively disappear as obesity sets in *Feuerer et al., 2009*. It is also known that TNFα levels in VAT increase with obesity (*Hotamisligil et al., 1993*), and that TNFα is detrimental to Treg function (*Nie et al., 2013*). Thus, we wondered whether the effect of surrogate peptide injection in this context would be enhanced by combining it with contemporaneous blockade of TNFα signaling. To this end, we transferred vTreg53 TCR-tg Treg cells into CD45.1+ B6 mice that had been treated with either anti-TNFα or isotype-control IgG antibodies and fed a HFD for six weeks; and then immunized the recipient mice with CFA/Fat1562 *s.c.*, and *i.p.*-boosted the response with IFA/Fat1562 (*Figure 7G*). As expected, Fat1562 immunization or anti-

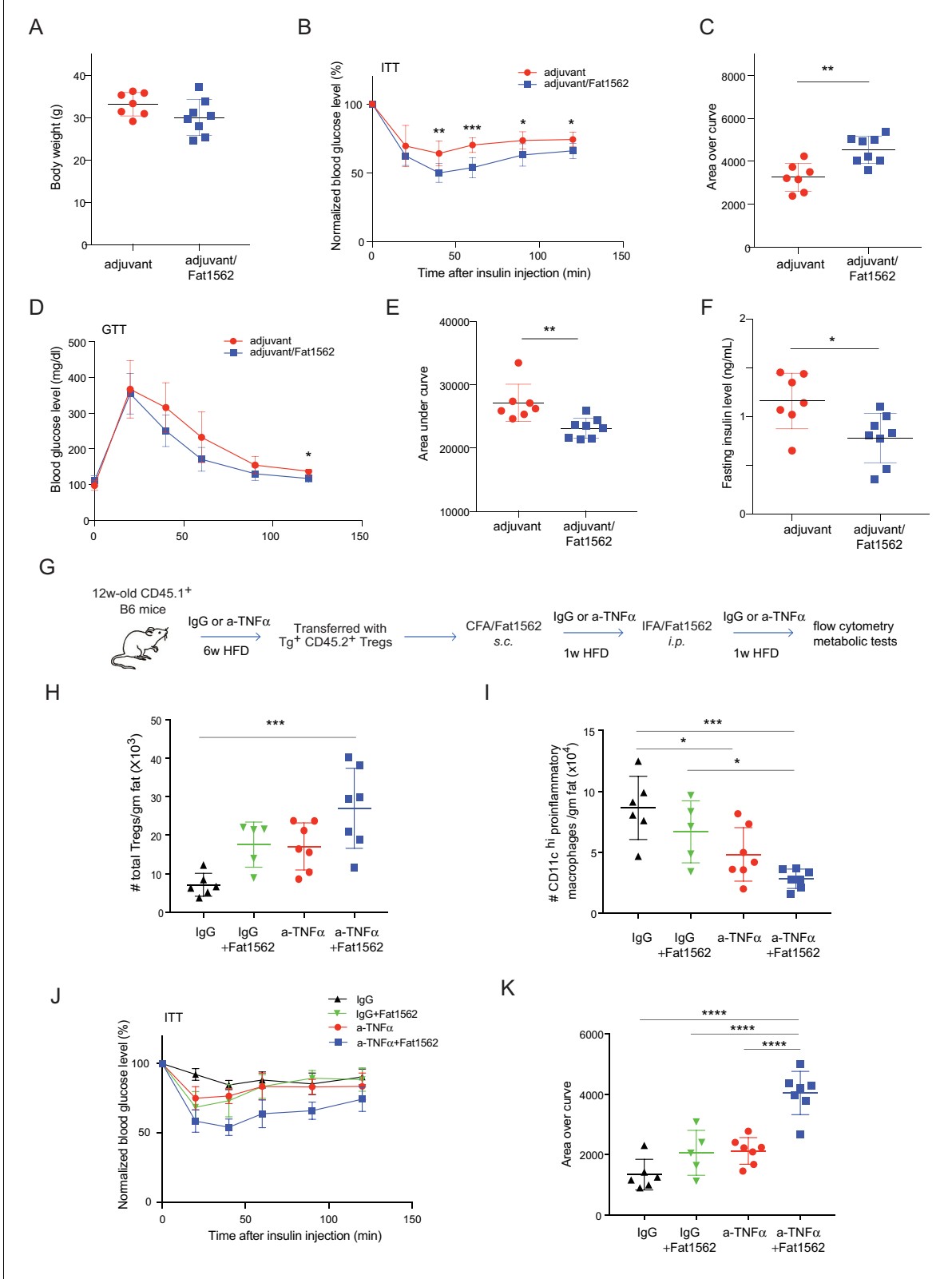

**Figure 7.** Immunization with Fat1562 following vTreg53 Treg transfer ameliorates HFD-induced insulin resistance. (**A–F**) Metabolic indices of mice treated as in *Figure 6A* (n ≥ 7). (**A**) Body weight. (**B**) Insulin tolerance test (ITT). (**C**) Area over curve for ITT analysis. (**D**) Glucose tolerance test (GTT). (**E**) Area under curve for GTT analysis. (**F**) Plasma insulin levels after 6 hr of fasting. Data shown are representative of at least two independent experiments.

*Figure 7 continued on next page*

*Figure 7 continued*

(**G**) Scheme of the anti-TNFα experiment. (**H**) Number of total Treg cells in VAT (n ≥ 5). (**I**) Number of CD11c[hi] inflammatory macrophages in VAT (n ≥ 5). (**J**) ITT (n ≥ 5). (**K**) Area over curve for ITT analysis (n ≥ 5). Data are mean ± SD.

The online version of this article includes the following figure supplement(s) for figure 7:

**Figure supplement 1.** Absolute blood glucose levels.

TNFα alone moderately increased the numbers of VAT-Treg cells (*Figure 7H*), reduced the number of CD11c[hi] pro-inflammatory macrophages (*Figure 7I*), and slightly improved insulin sensitivity in the recipient mice (*Figure 7J and K*). Co-administration of Fat1562 and anti-TNFα strikingly increased VAT-Treg cells, suppressed pro-inflammatory macrophages, and strongly promoted insulin sensitivity (*Figure 7H–K*). Collectively, these results demonstrated that Fat1562 could expand transferred vTreg53 Treg cells, dampen VAT inflammation, and improve insulin sensitivity in mice fed a HFD.

## Discussion

The small subset of Foxp3[+]CD4[+] Treg cells, comprising up to 2% of peripheral blood lymphocytes, plays a decisive role in maintaining immune homeostasis and reducing excessive immune activation (*Sakaguchi et al., 2008*), as well as several aspects of non-immune-tissue homeostasis. Importantly, several studies demonstrated that TCR expression is essential for Treg cell function (*Levine et al., 2014*; *Schmidt et al., 2015*) and that the TCR repertoire of these cells can recognize self- as well as foreign peptides (*Jordan et al., 2001*; *Lee et al., 2012*; *Pacholczyk et al., 2007*). Furthermore, TCR stimulation shapes various Treg-cell phenotypes (*Zemmour et al., 2018*). There is still, however, a paucity of methods to identify agonist Treg peptide ligands. Here, we took advantage of an unbiased peptide-library screen to identify potent agonists for a previously characterized, VAT-localized, Treg clone, vTreg53. Our goal was to elucidate whether the induction of an antigen-specific expansion of a particular population of VAT-Treg cells can ameliorate metabolic indices and improve insulin sensitivity in the context of a high-fat diet. Our results suggest that SPs can promote clonal Treg-cell proliferation without compromising key features of the VAT-Treg transcriptome, while suppressing HFD-induced VAT inflammation and improving insulin sensitivity.

In cancer immunotherapy, a major effort has been devoted to the identification of (neo)antigens that drive CD8[+] or CD4[+] anti-tumor responses. The success of this approach has been largely facilitated by deep sequencing analysis of the tumor cells' transcriptome coupled to the development of algorithms to identify mutations in protein-coding regions and the likelihood that these neoantigens are displayed by a given MHC allele (*Schumacher et al., 2019*). Unlike neoantigens, which can be readily identified by DNA sequencing, any wild-type peptide with suitable anchor residues may be presented by a given MHC molecule. The identification of self-antigens poses therefore a challenge that has been difficult to overcome. Current approaches to deorphanize Treg-cell TCRs have relied on the focused screening of organ-specific cDNA libraries (*Wang et al., 2004*), the isolation of thymus-derived MHCII-eluted peptides (*Stadinski et al., 2019*) and lastly, on the screening of organ-specific, Aire-dependent, proteins (*Leonard et al., 2017*). While these approaches have successfully revealed Treg-specific wild-type antigens, they also impose limitations regarding the tissue of origin or rely on the analysis of a relatively large number of APCs. For example, the limited number of MHCII-positive cells present in VAT currently precludes analysis of the A[b]-peptidome by mass spectrometry. A quick analysis of the mouse proteome reveals that more than half a million peptides, based solely on the presence of favorable anchor residues, can be presented by A[b]. Without a well-defined criterion to restrict the search of self-peptides to a reduced, tissue-specific protein-subset, it becomes nearly impossible to individually test the entire A[b]-peptidome for T cell responses.

Recent work suggests that the TCR has a rather reduced cross-reactivity and shows, instead, a poly-specific response (*Birnbaum et al., 2014*; *Wucherpfennig et al., 2007*). In other words, the TCR mostly recognizes a small, closely related, number of peptides. We have thus hypothesized that these newly identified SPs could guide the identification of self-peptides, similarly to what has been described previously (*Birnbaum et al., 2014*; *Gee et al., 2018*). However, the SPs identified using the vTreg53 TCR failed to fully align with a peptide region from the mouse proteome and, despite extensive effort, we could not deorphanize the vTreg53 TCR with an endogenous antigen.

Furthermore, candidate wild-type peptides relatively similar to Fat7 or Fat15 failed to activate vTreg53 cells. There may be several reasons behind our inability to identify an endogenous ligand, for example: (a) the preferred anchor residues for $A^b$ are ill-defined; (b) the theoretical diversity of a 13-mer peptide library exceeds by almost a 1000-fold the size of the yeast peptide-MHC library; (c) the depth of the $A^b$ peptide-presenting groove suggests that the TCR may have limited access to the peptide, which may impose an even greater peptide/TCR specificity for peptides presented by $A^b$; (d) register-shifting of the peptides presented by the engineered $A^b$ platform; (e) the affinity for self-peptides may fall below the detection limit of the yeast selection approach, particularly, when taking into account the reduced avidity of the yeast-display system; (f) lastly, several reports have documented that strong agonists can be created by post-translational modifications such as trans-peptidation or deamidation (*Hanada et al., 2004*; *Vigneron et al., 2004*). These and other limitations may complicate the identification of self-peptides presented by the $A^b$ MHC and potentially by class II MHCs and Treg cell TCRs, in general.

Despite the limitations described above, we were able to identify potent SP agonists. Altered peptide ligands, or SPs, have long been used to elicit more potent T cell responses, and often represent a useful approach to circumvent poor TCR recognition due to weak peptide binding to MHC, multiple peptide registers, lower avidity or poor TCR/pMHC affinity (*Sloan-Lancaster and Allen, 1996*). Furthermore, SPs have proven to be a valuable tool to explore key aspects of peptide recognition and the T cell response (*Evavold and Allen, 1991*; *Evavold et al., 1995*). More recently, SPs were shown to drastically suppress autoimmune responses during experimental autoimmune encephalomyelitis, a murine model for multiple sclerosis (*Saligrama et al., 2019*).

In the case of vTreg53, we identified several structurally related peptides that appeared to elicit a wide range of signaling potencies. Analysis of the Treg cells activated by the strongest SP, Fat1562, revealed two critical features. First, peptide-induced expansion of the Treg population was restricted to those cells displaying a very small number of TCRs. Whether this restriction is a property of the peptide used or an indication of a limited, highly peptide-specific VAT-Treg TCR repertoire remains to be investigated. Second, the transcriptome of Fat1562-activated Treg cells maintained the typical VAT-Treg signature, namely an enrichment of genes involved in lipid metabolism and an increase in *Pparg* mRNA levels (*Cipolletta et al., 2012*; *Feuerer et al., 2009*). Importantly, immunization with Fat1562 led to a significant expansion of adoptively transferred vTreg53 cells, accompanied by reduced VAT inflammation and improved systemic insulin sensitivity in the context of HFD. Previous studies showed that long-term HFD feeding is able to induce a VAT microenvironment that is highly toxic to VAT-Treg cells, partially due to increased level of TNFα (*Cipolletta et al., 2015*). We therefore reasoned that combination of anti-TNFα and Fat1562 immunization should be able to unleash the full potential of VAT-Treg cells. Indeed, co-administration of Fat1562 with anti-TNFα led to a striking increase of VAT-Treg cells and a significant improvement in insulin sensitivity in mice with severe obesity.

There has been substantial interest in harnessing the power of Treg cells to therapeutic ends, in particular for the amelioration of graft-versus-host, autoimmune or inflammatory diseases (*Sakaguchi et al., 2008*). A major obstacle relates, however, to the TCR de-orphanization of Treg clones. While polyclonal stimulation of Treg cells readily induces broad clonal expansion, antigen-specific stimulation of tissue-resident Treg cells has been found to be more effective at suppressing immune responses. In murine models of type-1 diabetes, antigen stimulation of Treg cells performed better at ameliorating diabetes progression than polyclonal Treg cells (*Tang et al., 2004*; *Yamazaki et al., 2006*), and similar results were found in humanized-mouse transplant models (*Putnam et al., 2013*). Moreover, recent data on the phenotypic distinction of non-lymphoid-tissue Treg cells argue that they will be a more effective source then their lymphoid-organ counterparts (*Panduro et al., 2016*). By analogy with the recent developments in TCR-based cancer immunotherapy, the potential to deorphanize disease-related Treg cell TCRs is likely to spur a greater focus on the discovery of disease-specific Treg populations. Surrogate peptides may offer a simpler and viable alternative to self-antigens in promoting local, tissue-specific, Treg expansion and immunosuppressive activity.

# Materials and methods

## Key resources table

| Reagent type (species) or resource | Designation | Source or reference | Identifiers | Additional information |
|---|---|---|---|---|
| Genetic reagent (*Mus. musculus*) | C57BL/6(J) | Jackson lab | Stock No: 000664 | |
| Genetic reagent (*Mus. musculus*) | CD45.1. C57BL/6 | Jackson lab | Stock No: 002014 | |
| Genetic reagent (*Mus. musculus*) | vTreg53 TCR-tg | *Li et al., 2018* | | |
| Genetic reagent (*Mus. musculus*) | Foxp3-GFP | Gift of Vijay. Kuchroo, (Brigham and Women's Hospital) | | |
| Cell line (*Homo sapiens*) | Jurkat E6.1 | ATCC | Cat# TIB-152 RRID:CVCL_0367 | |
| Cell line (*Homo sapiens*) | Lenti-X 293T | Takara Bio | Cat# 632180 | |
| Cell line (*Homo sapiens*) | K-562 | ATCC | Cat# CCL-243, RRID:CVCL_0004 | |
| Antibody | anti-CD45.1 (clone A20) | Biolegend | Cat# 110724 RRID:AB_493733 | FACS (1:100) |
| Antibody | anti-CD45.2 (clone 104) | Biolegend | Cat# 109824 RRID:AB_830789 | FACS (1:100) |
| Antibody | anti-CD3 (clone 17A2) | Biolegend | Cat#: 100214 RRID:AB_493645 | FACS (1:100) |
| Antibody | anti-CD4 (clone GK1.5) | Biolegend | Cat#: 100422 RRID:AB_312707 | FACS (1:100) |
| Antibody | anti-TCR V$\alpha$2 (clone B20.1) | Biolegend | Cat#: 127806 RRID:AB_1134188 | FACS (1:100) |
| Antibody | anti-CD44 (clone IM7) | Biolegend | Cat#: 103032 AB_2076204 | FACS (1:100) |
| Antibody | anti-CD62L (clone MEL-14) | Biolegend | Cat#: 104418 RRID:AB_313103 | FACS (1:100) |
| Antibody | anti-CD45 (clone 30-F11) | Biolegend | Cat#: 103126 RRID:AB_493535 | FACS (1:100) |
| Antibody | anti-CD11b (clone M1/70) | Biolegend | Cat#: 101228 RRID:AB_893232 | FACS (1:100) |
| Antibody | anti-CD11c (clone N418) | Biolegend | Cat#: 117318 RRID:AB_493568 | FACS (1:100) |
| Antibody | anti-F4/80 (clone BM8) | Biolegend | Cat#: 123116 RRID:AB_893481 | FACS (1:100) |
| Antibody | anti-c-Myc Alexa Fluor 488 | Cell Signaling Technology | Cat# 9402 RRID:AB_2151827 | FACS (1:100) |
| Antibody | anti-CD69 (clone FN40) | Biolegend | Cat# 104508, RRID:AB_313111 | FACS (1:100) |
| Antibody | anti-CD25 (clone M-A25) | Biolegend | Cat# 356110, RRID:AB_2561977 | FACS (1:100) |
| Recombinant protein | Streptavidin APC | Biolegend | Cat#: 405243 | FACS (40–400 nM) |
| Antibody | anti-TCR V$\beta$4 (KT4) | BD Biosciences | Cat#: 553366 RRID:AB_394812 | FACS (1:100) |
| Antibody | anti-Siglec-F (E50-2440) | BD Biosciences | Cat#: 552126 RRID:AB_394341 | FACS (1:100) |
| Antibody | anti-Foxp3 (clone FJK-16s) | ThermoFisher | Cat#: 17-5773-82 RRID:AB_469457 | FACS (1:100) |

*Continued on next page*

*Continued*

| Reagent type (species) or resource | Designation | Source or reference | Identifiers | Additional information |
|---|---|---|---|---|
| Antibody | anti-TNFα (clone XT3.11) | BioXCell | Cat#: BE0058 RRID:AB_1107764 | In vivo 10 µg/g |
| antibody | isotype control IgG (clone HRPN) | BioXCell | Cat#: BE0088 RRID:AB_1107775 | In vivo 10 µg/g |
| Commercial assay or kit | Streptavidin microbeads | Miltenyi | Cat#: 130-048-101 | 50 µl/sample |
| Commercial assay or kit | Foxp3/Transcription Factor Staining Buffer Set | ThermoFisher | Cat#: 00-5523-00 | |
| Commercial assay or kit | CellTrace Violet dye | ThermoFisher | Cat#: C34557 | |
| Commercial assay or kit | Ultra Sensitive Mouse Insulin ELISA Kit | Crystal Chem | Cat#: 90080 | |
| Chemical compound, drug | D-(+)-Glucose | Sigma Aldrich | Cat#: G8270-1KG | |
| Chemical compound, drug | Humulin R | Lilly | Cat#: U-100 | |
| Chemical compound, drug | Freund's Adjuvant, Complete | Sigma | Cat#: F5881 | |
| Chemical compound, drug | Freund's Adjuvant, Incomplete | Sigma | Cat#: F5506 | |
| Software, algorithm | Multiplot Studio | Genepattern | https://www.genepattern.org/modules/docs/Multiplot/2 | |
| Software, algorithm | GSEA | Broad Institute | https://www.gsea-msigdb.org/gsea/index.jsp | |
| Software, algorithm | V-QUEST | IMGT | http://www.imgt.org/IMGT_vquest/user_guide | |

## Cell line culture and peptide stimulation

Cell lines were kept in a humidified incubator at 37°C with 5% $CO_2$ unless otherwise denoted. Jurkat T cells (derived from a male with acute T cell leukemia; ATCC TIB-152) and K562 cells (ATCC CCL-243) were cultured in RPMI supplemented with 2 mM glutamax (Invitrogen), 10% FBS (Sigma),10 mM HEPES pH 8.0 (ThermoFisher), 1 mM Sodium Pyruvate (Gibco) and 50 U/ml penicillin and streptomycin (ThermoFisher). Validation of T cell lines was performed by staining with known markers pre- and post- transfection or transduction. HEK293T (Lenti-X; Takara Bio USA, #632180) cells (female derived kidney cell line) were grown in DMEM complete media (ThermoFisher) supplemented with 10% FBS, 2 mM L-glutamine, and 50 U/ml of penicillin and streptomycin. Cells were tested for mycoplasma contamination using the MycoAlert Mycoplasma Detection kit (Lonza) and found to be mycoplasma-negative. For peptide assays, Jurkat T cells were rested overnight or for 2–3 hr in fresh RPMI complete. Cells were co-cultured for 16 to 48 hr with surrogate APCs: K562 transduced with $A^b$. After overnight incubation, cells were washed in MACS buffer and stained with anti-TCR, $A^b$, CD69 and CD25. EBY100 yeast cells were cultured in sterile filtered standard YPD (Bacto Peptone, glucose and yeast extract). Transformed yeast were grown in SDCAA (yeast nitrogen base, casamino acids -ADE, -URA, -TRP, D-glucose and buffered to pH 4.5 using sodium citrate and citric acid monohydrate) and induced in SGCAA where galactose replaces D-glucose. EBY100 cells are grown at 30°C when cultured in SDCAA, or at 20°C when cultured in SGCAA at atmospheric $CO_2$ in an orbital shaker set to an agitation speed of 250 rpm.

## Protein expression

High Five cells (Tni; Expression Systems) were grown in Insect X-press media (Lonza) or ESF 921 media (Expression Systems) with a final concentration of 10 mg/L of gentamicin sulfate (ThermoFisher) at 27°C at atmospheric $CO_2$. SF9 cells are grown in SF900-III or -II serum-free media (ThermoFisher) with 10% FBS and final concentration 10 mg/L of gentamicin sulfate and 2 mM glutamax at 27°C at atmospheric $CO_2$. P1 or P2 virus was used to infect volumes of 1–3 L of Hi5 cells at ~2 × $10^6$

cells/ml. Supernatant was harvested 2–3 days post-infection and spun down at 8000 rpm for 15 min. The supernatant containing expressed protein was treated to 100 mM Tris pH 8.0, 1 mM NiCl$_2$, and 5 mM CaCl$_2$ to precipitate contaminants. The supernatant and precipitate mixture was spun down at 8000 rpm for 20 min at 4°C to remove precipitate. The supernatant was incubated with Ni-NTA resin (QIAGEN) for >3 hr at room temperature. Ni-NTA beads were collected and washed in a Buchner funnel with 20 mM imidazole in 1 x HBS pH 7.2 and eluted with 200 mM imidazole in 1x HBS pH 7.2. Protein was concentrated in a 10 kDa filter (Millipore, UFC903024) to ~1 mL. When appropriate, protein was biotinylated with BirA ligase, 100 μM biotin, 40 mM Bicine pH 8.3, 10 mM ATP, and 10 mM Magnesium Acetate at 4°C overnight. All proteins were further purified by size-exclusion chromatography using Superdex Increase S200 or S75, as appropriate (GE Healthcare). All proteins for in vivo studies were cleared of endotoxin. Final endotoxin levels were determined using a chromogenic endotoxin quantitation kit (Thermo Fisher) and never greater than 1 Endotoxin Unit/mg of purified protein.

## Development and selection of peptide-A$^b$ yeast libraries

To obtain a functional A$^b$ yeast display platform, a 13mer-peptide sequence linked the β1-α1 expressed on the surface of the *S. cerevisiae* strain EBY100 as an N-terminal fusion to Aga2 using the pYAL vector was evolved by error-prone PCR. The GeneMorph II random mutagenesis kit (Agilent) was used to introduce between 3 to 5 mutations per one kbp. Mutation frequency and overall integrity of the A$^b$ expression gene was confirmed by DNA sequencing following ITA cloning and DNA miniprep from randomly selected single-colonies (up to 50 colonies were used). The final error-product was amplified to generate 50 μg of insert DNA using high-fidelity DNA polymerase. A$^b$ libraries were created by electroporation of chemically competent EBY100 with mutagenized insert and 10 μg of linearized pYAL vector. Peptide-A$^b$ selections were performed as described below. Functional surface expression was obtained following two single-point mutations in the β-chain: His46Tyr and Leu67Val. A$^b$-peptide yeast display libraries were created as previously described (*Birnbaum et al., 2014*). Briefly, oligos allowing all 20 amino acids via NNK codons at all peptide positions except the defined anchors (as shown in *Figure 1A*) were used to generate a peptide library. The libraries allowed for limited diversity at P1, P4, P6 and P9 anchor positions to maximize the number of correctly folded pMHC clones on the surface of yeast. pMHC libraries with up to $2 \times 10^9$ transformants as determined by colony counting after serial dilution were generated by electroporation of chemically competent EBY-100 cells via homologous recombination with an insert composed by the peptide library fused to the β1(H46Y, L67V)-α1-c-myc and a linearized pYal plasmid. Three independent libraries were made, mixed, and screened by the TCRs here described as an attempt to reduce potential PCR biases.

Yeast were passaged in SDCAA and induced with SGCAA and selected with streptavidin (SA) - coated magnetic MACS beads (Miltenyi) coated with biotinylated TCR. The number of yeast used for each round of selection was ~10 x the diversity from the previous selection step. In the first round of yeast selection from the naïve library, yeast corresponding to 2-5x library diversity were incubated on a rotator at 4°C for 1 hr in 10 mL of ice cold PBS/0.5% bovine serum albumin and 1 mM EDTA (PBE) with 250 μl of SA beads. Yeast-bead mixture was negatively selected using an LS Column (Miltenyi) attached to a magnetic stand (Miltenyi) and washed three times with PBE. The flow through was further incubated with 400 μl SA beads preincubated with 400 nM of TCR for 3 hr at 4°C on a rotator. The yeast were washed and pelleted at 3500 g for 2 min prior to positive selection using an LS column. The yeast eluted from the column were grown in 3 mL of SDCAA until the OD >5, which usually corresponded to 36–48 hr. For rounds 2 and 3 of selection,~$3 \times 10^7$ (corresponding to 3 ml at OD of 1) yeast were induced in SGCAA for 2 days at 20°C, at which point OD was found to be >5. After confirming c-myc tag expression by flow cytometry, $10^7$ yeast were incubated with 50 μl of SA-beads for 1 hr for the negative selection followed by 50 μl of TCR coated beads for 3 hr in 500 μl of PBE. All incubations were performed at 4°C and TCR concentration was kept at 400 nM. The fourth round of selection was similar to round 2 and 3, while substituting SA-magnetic beads for SA alone. For the negative selection step, yeast were incubated with 400 nM streptavidin-APC (SA-APC) in 500 μl for 1 hr at 4°C, followed by a 20 min incubation with 50 μl of microbeads coated with anti-647 (Miltenyi). For the subsequent positive selection step, yeast were incubated with 400 nM TCR tetramer for 3 hr at 4°C followed by 20 min with anti-alexa647 magnetic beads. Peptide-A$^b$ surface expression was monitored with anti-c-myc (Cell Signaling) antibody staining. All selections rounds

and naïve libraries were kept for up to 2 months at 4°C for subsequent DNA extraction, induction and/or TCR-SA tetramer staining.

## Deep sequencing

DNA was isolated from each round of selection using the a Zymoprep II Kit (Zymo Research) as per manufacturer's instructions. Briefly, $5 \times 10^7$ yeast from all rounds were lysed and DNA was extracted and eluted in 10 mM Tris pH 8.0 and kept at 4°C for further processing. To amplify the DNA region corresponding to the peptide sequences, a forward oligo with unique barcodes and a random 8-mer sequence was used together with a reverse oligo designed to anneal the linker connecting the peptide region to the β1 domain of $A^b$. DNA was amplified by PCR for up to 15 cycles. PCR products were purified after agarose gel electrophoresis and later re-amplified by PCR for 15 cycles to include the standard Illumina adaptor sequences. All amplification steps were performed using Phusion DNA Polymerase (NEB). The final amplified library was purified following agarose gel electrophoresis and quantified using nanodrop. Integrity of the amplified DNA product was confirmed by BioAnalyzer (Agilent Genomics). The library was deep sequenced using the Illumina Miseq sequencer and a V2 (2 × 150) or V3 kit.

## Analysis of deep sequencing data

Paired-end reads were identified and selected for further analysis from the deep sequencing using PandaSeq (*Masella et al., 2012*) after which the paired-end reads were parsed according to the barcodes using Geneious V6. Sequences with frameshifts or stop codons were removed from further analysis. Sequences were processed to quantify the number of identical peptide sequences using custom Perl scripts and shell commands.

## Surface Plasmon resonance

A GE Biacore T100 was used to measure the $K_D$ by equilibrium methods. Approximately 200 resonance units (RU) of human CD45, PD-1 or mouse PD-1 was captured on a SA-chip (GE Healthcare), including a reference channel of an unrelated protein. SPR runs were performed in HBS-P+ (GE Healthcare). All measurements were made with 2-fold serial dilutions using 45–60 s association (at 30 µl/min) followed by >240 s dissociation (30 µl/min) at 25°C. Regeneration was performed using HBS-P+, 2 M MgCl$_2$ or 0.1 M Glycine, pH 3.1 for 15–20 s at 50 µl/min. Measurement of titrations at equilibrium were used to determine the $K_D$ using Biacore Analysis Software (GE Healthcare). All measurements were repeated twice.

## Mice

Male mice of various ages were produced in our specific-pathogen-free facilities at Harvard Medical School, and were fed a chow diet (no. 5058, Lab Diet Picolab Mouse Diet 20), or HFD (60 kcal% Fat, Research Diets, D12492i). B6.CD45.1$^+$ mice were purchased from the Jackson Laboratory. *Foxp3-GFP* mice were obtained from Dr. V. Kuchroo. vTreg53 TCR-tg mice were generated based on one of the expanded VAT-Treg clones, as previously described (*Li et al., 2018*), and were crossed with *Foxp3-GFP$^{KI/KI}$* females. Tg$^+$ Foxp3-GFP$^{KI/y}$ male progeny were used for experiments. In all cases, the *Tcra* plus *Tcrb* transgene was maintained in the heterozygous state. All experiments were performed using littermate controls and following animal protocols approved by the HMS Institutional Animal Use and Care Committee (protocol IS00001257).

## Cell isolation and flow cytometry

Mice were asphyxiated with CO$_2$. Epididymal VAT was excised, minced and digested for 20 min with 1.5 mg/ml collagenase type II (Sigma) in Dulbecco's Modified Eagle's Medium (DMEM) supplemented with 2% fetal calf serum (FCS) in a 37°C water bath with shaking. The digested materials were filtered through a sieve and then a 40 µm nylon cell strainer. The stromal vascular fraction was collected after centrifugation at 450 g for 10 min. For surface markers, cells were stained with anti-CD45.1 (A20), -CD45.2 (104), -CD3 (17A2), -CD4 (GK1.5), -TCR Vα2 (B20.1), -CD44 (IM7), -CD62L (MEL-14), -CD45 (30-F11), -CD11b (M1/70), -CD11c (N418), -F4/80 (BM8) mAbs (all from Biolegend), and anti-TCR Vβ4 (KT4), anti-Siglec-F (E50-2440) mAbs (both from BD Biosciences), as detailed in experiment descriptions. Foxp3 staining was performed using the Foxp3/Transcription Factor

Staining Buffer Set and anti-Foxp3 (FJK-16s) mAb (both from eBioscience). Cells were acquired with an LSRII flow cytometer (BD Biosciences), an Accuri (BD Biosciences) or Cytoflex (Beckman Coulter). Propidium Iodide was used to distinguish live from dead cells. Assays were performed in biological duplicates or triplicates as described in the figure legends. Data were analyzed using FlowJo.

## Primary Treg cultures and peptide stimulation

Tregs from pooled LNs of Tg[+] Foxp3-GFP[KI/y] mice were labeled with CellTrace Violet dye (Thermo-Fisher Scientific), cultured in complete RPMI medium in the presence of 100 U/ml recombinant mouse IL-2 (peprotech), and stimulated by syngeneic APCs loaded with the indicated concentrations of different peptides. 3 days later, the proliferation and activation of clonotype[+] Treg cells was determined by flow cytometric analysis of the dilution of the CellTrace Violet dye, upregulation of CD44 expression, and downregulation of CD62L levels.

## Treg transfers and peptide immunization

To assess early Treg proliferation following peptide stimulation in vivo, 0.4 million Treg cells were sorted from pooled Spl and LNs of 6-8wk-old male CD45.2[+] Tg[+] Foxp3-GFP[KI/y] mice by Moflo, labeled by the CellTrace Violet dye (ThermoFisher Scientific), and were i.v.-injected into 6-8wk old male B6.CD45.1[+] mice. The next day, the recipient mice were injected with 200 µg peptides emulsified in CFA (CFA/peptides) s.c. on the lower back. Proliferation of clonotype[+] Tregs in the spleen was determined three days later. To examine the expansion of VAT-Treg cells by peptide immunization, we sorted 0.4 million Treg cells from pooled Spl/LNs of 6-8wk-old male CD45.2[+] Tg[+] Foxp3-GFP[KI/y] mice and transferred them i.v. into 6-8wk old male B6.CD45.1[+] mice. Six weeks later, the recipient mice were immunized with 200 µg CFA/peptides s.c. on the lower back, and one week later were boosted with 100 µg IFA/peptides i.p. After another week, clonotype[+] CD45.2[+] Treg cells from the Spl/LNs or VAT of recipient mice were analyzed by flow cytometry and double-sorted for ultra-low-input RNA-seq. Lastly, for transfer into mice fed a HFD, 0.4 million Treg cells were sorted from pooled Spl/LNs of 6-8wk-old male CD45.2[+] Tg[+] Foxp3-GFP[KI/y] mice and transferred i.v. into 12-week-old B6.CD45.1[+] mice that had been fed a HFD for 2 weeks. The next day, recipient mice were immunized with 200 µg CFA/peptides s.c. on the lower back, and one week later were boosted with 100 µg IFA/peptides i.p. Mice were maintained on HFD for the duration of the procedure. One week after the IFA/peptides injection, metabolic parameters (GTT, ITT, fasting plasma insulin levels) and the composition of different immunocytes were determined.

## Anti-TNFα treatment

12-week-old B6.CD45.1[+] mice were fed with HFD and injected intraperitoneally with 10 µg/g body weight anti-TNFα (XT3.11, BioXCell) or isotype control IgG (HRPN, BioXCell) twice a week for 6 weeks, transferred with 0.4 million Treg cells from pooled Spl/LNs of 6-8wk-old male CD45.2[+] Tg[+] Foxp3-GFP[KI/y] mice. The next day, recipient mice were immunized with 200 µg CFA/peptides s.c. on the lower back, and one week later were boosted with 100 µg IFA/peptides i.p. Mice were maintained on HFD and continuously treated with anti-TNFα or IgG twice a week for the duration of the entire experiment. One week after the IFA/peptides injection, ITT (0.75unit/kg insulin) was performed and the composition of different immunocytes were determined.

## Ultra-low-input RNA-seq library preparation and data analysis

6–8 week-old B6.CD45.1[+] mice were transferred with Tg[+] Treg cells from Spl/LNs, and six weeks later were primed with 200 µg CFA/peptides s.c. and boosted with 100 µg IFA/peptides i.p. as described in the 'Treg transfer and peptide immunization' section. Biological replicates (n $\geq$ 2) of 1000 clonotype[+] CD45.2[+] Treg cells from the Spl/LNs or VAT of recipient mice were double-sorted by Moflo directly into 5чl lysis buffer (TCL Buffer (Qiagen) with 1% 2-Mercaptoethanol), Smart-seq2 libraries were prepared as previously described (*Picelli et al., 2014*) with slight modifications. Briefly, total RNA was captured and purified on RNAClean XP beads (Beckman Coulter). Polyadenylated mRNA was then selected using an anchored oligo(dT) primer (5'–AAGCAGTGGTATCAACGCAGAG TACT30VN-3) and converted to cDNA via reverse transcription. First-strand cDNA was subjected to limited PCR amplification followed by Tn5 transposon-based fragmentation using the Nextera XT DNA Library Preparation Kit (Illumina). Samples were then PCR amplified for 18 cycles using

barcoded primers such that each sample carried a specific combination of eight-base Illumina P5 and P7 barcodes and were pooled together prior to Smart sequencing. Smart-seq paired-end sequencing was performed on an Illumina NextSeq500 using 2 × 38 bp reads with no further trimming. Transcripts were quantified by the Broad Technology Labs computational pipeline with Cuffquant version 2.2.1 (*Trapnell et al., 2012*). Normalized reads were further filtered by minimal expression (>10) and coefficient of variation (<0.7), and then analyzed by Multiplot Studio in the GenePattern software package. Pathway enrichment analysis was done using GSEA.

## Single-cell TCR sequencing analysis

Foxp3$^+$CD4$^+$ TCRβ$^+$ Fat1562/A$^b$-PE$^+$ Treg cells from VAT or pooled Spl/LN of 16–20 week-old *Foxp3-GFP* mice were first sorted in bulk before resorting as individual cells using a MoFlo (Beckman Coulter) into wells of 96-well PCR plates containing the reverse transcriptase reaction mix, and cDNA was prepared as described previously (*Burzyn et al., 2013*). Extreme caution was taken to minimize contamination of wells. Preparation of the reverse transcriptase reaction mix, synthesis of cDNA and performance of the first round of PCR were done in a different building from the second-round PCR and final PCR product isolation. At least 2 columns of every plate were left blank to serve as negative controls to monitor for contamination. Resulting cDNA (1.5 μl) from each cell was split to perform multiplex nested PCR reactions to amplify the corresponding CDR3α- and CDR3β-encoding transcripts using the protocol and CDR3β primers published in *Burzyn et al., 2013*. Aliquots of the PCR products were visualized on a 1.5% agarose gel. Samples containing PCR products encoding both the TCRα and TCRβ chains were cleaned up using ExoSAP-IT For PCR Clean-Up (Affymetrix) per the manufacturer's instructions and were subjected to automated sequencing at the Dana-Farber/Harvard Cancer Center High-Throughput Sequencing Core. Raw sequencing files were filtered for sequence quality, processed in automated fashion, and parsed using IMGT/V-QUEST (*Brochet et al., 2008*). Only sequences that produced functional in-frame rearrangements of both the TCRα and TCRβ chains for a given clone were included in the analysis.

## Metabolic studies

For GTT, mice were fasted for 15 hr overnight, weighed, and tested for fasting blood-glucose concentrations. Glucose (2.0 g per kg body weight) was administered by *i.p.* injection. Blood-glucose levels were measured before and 20, 40, 60, 90, and 120 min after glucose injection. For the ITT, mice were fasted for 6 hr before being injected i.p. with insulin (0.6 U per kg body weight, Humulin R, Lilly). Plasma was collected for fasting insulin concentrations (Crystal Chem Ultra Sensitive Mouse Insulin ELISA Kit). Blood-glucose levels were measured before and 20, 40, 60, 90, and 120 min after insulin injection, and were normalized to the blood-glucose levels before insulin injection. The area under the curve (for GTT) or area over the curve (for ITT) were calculated using GraphPad Prism 7.0.

## Statistics

All figures are representative of at least two (in vivo) or at least three (in vitro) independent experiments unless otherwise noted. Sample sizes were chosen based on our experience with similar experiments (a minimum of 4–6 mice for animal studies, or 2–4 biological replicas for in vitro assays). Unless stated otherwise, significance was assessed by Student's t test or ANOVA using GraphPad Prism 7.0 (two-tailed, unpaired or paired t-test depending on the experiment). To determine the enrichment of gene signatures in RNA-seq datasets, we used a Chi-square test. *: $p < 0.05$; **: $p < 0.01$; ***: $p < 0.001$; ****: $p < 0.0001$; ns: not significant. Data are represented as mean ± standard deviation (sd) unless otherwise stated.

## Data availability

Deep-sequencing data for the peptide-Ab yeast library screening and RNA-seq data have been deposited in GEO under accession codes GSE151070 and GSE150173. Custom Perl scripts for the processing of the deep sequencing data for the peptide-Ab is available from: *Fernandes, 2020* https://github.com/jlmendozabio/NGSpeptideprepandpred copy archived at https://github.com/elifesciences-publications/NGSpeptideprepandpred.

## Acknowledgements

We thank all members from DM and KCG laboratory for helpful discussions and reagents. We thank Dr. Naresha Saligrama and Dr. Marvin Gee for experimental advice and discussions. Wellcome Trust (WT101609MA to RAF) NIH (5R01AI103867), Howard Hughes Medical Institute, (UC4DK116264) and Mathers Foundation to KCG. NIH (2R01 DK092541) and the JPB Foundation to DM. CL was supported by a fellowship from the Cancer Research Institute.

## Additional information

### Funding

| Funder | Grant reference number | Author |
|---|---|---|
| Wellcome | WT101609MA | Ricardo A Fernandes |
| NIH Office of the Director | 5R01AI103867 | K Christopher Garcia |
| Howard Hughes Medical Institute | HHMI | K Christopher Garcia |
| G Harold and Leila Y. Mathers Foundation | | K Christopher Garcia |
| NIH Clinical Center | 2R01 DK092541 | Diane Mathis |
| JPB Foundation | | Diane Mathis |
| NIH Office of the Director | UC4DK116264 | K Christopher Garcia |

The funders had no role in study design, data collection and interpretation, or the decision to submit the work for publication.

### Author contributions

Ricardo A Fernandes, Conceptualization, Data curation, Formal analysis, Investigation, Methodology, Writing - original draft, Writing - review and editing; Chaoran Li, Conceptualization, Data curation, Formal analysis, Validation, Investigation, Methodology, Writing - original draft, Writing - review and editing; Gang Wang, Xinbo Yang, Formal analysis, Investigation, Methodology; Christina S Savvides, Formal analysis, Investigation; Caleb R Glassman, Shen Dong, Eric Luxenberg, Investigation, Methodology; Leah V Sibener, Resources, Investigation, Methodology; Michael E Birnbaum, Resources, Methodology; Christophe Benoist, Conceptualization, Resources, Investigation, Methodology; Diane Mathis, Conceptualization, Data curation, Supervision, Funding acquisition, Investigation, Writing - original draft, Writing - review and editing; K Christopher Garcia, Conceptualization, Data curation, Supervision, Funding acquisition, Writing - review and editing

### Author ORCIDs

Ricardo A Fernandes (iD) https://orcid.org/0000-0001-5343-3334
K Christopher Garcia (iD) https://orcid.org/0000-0001-9273-0278

### Ethics

Animal experimentation: This study was performed in strict accordance with the recommendations in the Guide for the Care and Use of Laboratory Animals of the National Institutes of Health and every effort was made to minimize suffering. All experiments were performed following animal protocols approved by the HMS Institutional Animal Use and Care Committee (protocol IS00001257).

### Decision letter and Author response

Decision letter https://doi.org/10.7554/eLife.58463.sa1
Author response https://doi.org/10.7554/eLife.58463.sa2

## Additional files

### Supplementary files

• Supplementary file 1. Deep-sequencing of peptide-Ab yeast library after selection with vTreg53 TCR.

• Transparent reporting form

### Data availability

Sequencing data for the peptide-Ab yeast library screening and RNA-seq data for VAT-Treg cells have been deposited in GEO under accession codes GSE151070 and GSE150173. Custom Perl scripts for the processing of the deep sequencing data for the peptide-Ab is available from: https://github.com/jlmendozabio/NGSpeptideprepandpred copy archived at https://github.com/elifesciences-publications/NGSpeptideprepandpred.

The following datasets were generated:

| Author(s) | Year | Dataset title | Dataset URL | Database and Identifier |
|---|---|---|---|---|
| Fernandes RA, Li C, Wang G, Yang X, Savvides CS, Glassman CR, Dong S, Luxemberg E, Sibener LV, Birnbaum ME, Benoist C, Mathis D, Garcia KC | 2020 | DNA sequencing for multiple rounds of the pMHC-yeast display selection for 2W, Yae and Fat TCR | https://www.ncbi.nlm.nih.gov/geo/query/acc.cgi?acc=GSE151070 | NCBI Gene Expression Omnibus, GSE151070 |
| Fernandes RA, Li C, Wang G, Yang X, Savvides CS, Glassman CR, Dong S, Luxemberg E, Sibener LV, Birnbaum ME, Benoist C, Mathis D, Garcia KC | 2020 | Transcriptional profiling of vTreg53 TCR transgenic Regulatory T (Treg) cells stimulated by agonist peptide | https://www.ncbi.nlm.nih.gov/geo/query/acc.cgi?acc=GSE150173 | NCBI Gene Expression Omnibus, GSE150173 |

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
