## [Decision Letter]

**Acceptance summary:**

Foxp3^+^CD4^+^ T regulatory (Treg) cells negatively regulate immune response and also act in non-lymphoid organs, where they control local immune responses and help maintaining tissue homeostasis. Among 'tissue-Tregs' those found in visceral adipose tissue (VAT) control local and systemic inflammatory and promote insulin sensitivity. Their actions depend on TCR recognition of peptide-MHC complexes; yet the degree of peptide specificity of Treg-cell function, and whether Treg ligands can be used to manipulate Treg cell biology are unknown. The present study uses yeast peptide display libraries to identify surrogate peptides that bind to class II MHC molecules and were able to stimulate Treg cells bearing a transgenic TCR isolated from a VAT resident Treg cell. These surrogate p-MHC complexes were capable of expanding transferred the corresponding TCR transgenic Tregs thereby protecting mice from some of the consequences of a high fat diet. Therefore, this study is important in that it suggests that antigen-specific targeting of VAT-localized Treg cells could eventually be a strategy for improving metabolic disease.

**Decision letter after peer review:**

Thank you for submitting your article "Discovery of surrogate agonists for visceral fat Treg cells that modulate metabolic indices in vivo" for consideration by *eLife*. Your article has been reviewed by three peer reviewers, and the evaluation has been overseen by a Reviewing Editor and Tadatsugu Taniguchi as the Senior Editor. The reviewers have opted to remain anonymous.

The reviewers have discussed the reviews with one another and the Reviewing Editor has drafted this decision to help you prepare a revised submission.

Using yeast peptide display libraries focusing on I-Ab:peptide complexes, the authors identified surrogate peptides that did not match peptides from the mouse genome but were able to stimulate cells bearing a transgenic TCR isolated from a VAT resident Treg cell. These surrogate peptides were able to expand transferred TCR transgenic Tregs, thereby protecting mice from some of the consequences of a high fat diet. Although the study is novel and well designed , a few important issues have been raised by the 3 reviewers and need to be addressed before publication in *eLife*. More specifically, it should be make clear in the Abstract and title that the expanded VAT-Treg cells express a transgenic TCR, and a peptide-only control condition need to be provided in the TNFa blocking experiments.

Reviewer #1:

In this manuscript, the authors generated the first I-Ab:peptide yeast display library, and successfully identified agonist peptides for a TCR cloned from a VAT Treg cell.

Unfortunately, the identified antigens did not match any peptide from the mouse genome, such that the physiological antigens for VAT Tregs remain unknown.

The study nevertheless demonstrates feasibility of antigen-dependent expansion of VAT Tregs. Importantly, expanded Tregs retained a VAT transcriptomic profile and improved insulin sensitivity in a model of diet-induced insulin resistance.

Overall, the study is well designed and executed, and the manuscript is clearly written. The novelty of the approach will be of interest to the immunology community.

Reviewer #2:

In recent years it has become clear that Tregs have a range of phenotypes and roles in different tissue sites. However how to selectively manipulate these cells remains highly difficult since while they may have some specific properties these tissue resident Tregs tend to share markers with either other Tregs or effector cells in the same site. Using a peptide display library the authors were able to find several surrogate peptides that were able to effectively stimulate cells bearing a TCR originally isolated from VAT resident Tregs. These SP were able to expand wildtype VAT Tregs in some mice and also were highly effective at the expansion of transferred TCR transgenic Tregs with the effect that they were able to protect from some of the consequences of a high fat diet.

The ability to selectively manipulate a tissue resident subpopulation of Tregs holds great promise for the development of organ specific treatments and the paper covers a topic of current interest and is generally well written. I do however have some comments on particular experiments and some concerns about the strength of the data including anti-TNFa.

1) Figure 4—figure supplement 1. The authors show some data from tetramer experiments. Even though only 3/10 mice responded it may be worth including graphs showing the number of tetramer positive cells from the different tissues. It is notable that the proportion of tetramer positive amongst Tregs is quite high in the VAT considering that no enrichment was performed, suggesting that at least in the representative mouse this worked rather well and may require more attention.

2) Figure 5B. I think a bit more effort can be put into the analysis of the differences between the groups. Using similar analysis to 5A were there any changes to the genes in the previously defined VAT UP and DOWN signature genes? What were the top differentially expressed genes and do any of them have known function in Tregs. How do the authors interpret the increased *Pparg* mRNA in the Spl/LN? Is this evidence of trafficking from the VAT? The authors note but do not show that other VAT signature genes may have increased.

3) Figure 7G-M: Since the authors have already demonstrated that Fat1562 alone is effective in this system it seems hard to say if anti-TNF is making a significant addition to its efficacy since there is no Fat1562 only group for comparison. The normalized blood glucose level of the anti-TNF + Fat1562 group in Figure 7L is not noticeably better than the Fat1562 alone group in 7B. Similarly, comparison of 7I and 6G shows no support for a clear contribution of anti-TNF. Direct comparison within one experiment might be able to show a contribution of anti-TNFa to the effect of Fat1562 but this has not been done and without it I'm not sure this data is meaningful.

4) It is also worth noting that similar data in Figures 6 and 7 are not presented consistently e.g. macrophage analysis is given as %CD11chi of F4/80 in 6H but #CD11chi/gram of fat in 7J. Raw numbers in 6E, x103 in 7I etc. These differences should be either justified or removed.

Reviewer #3:

In this manuscript the authors screen for peptides with the capacity to activate the vTreg53 TCR transgenic. The authors find several peptides that activate this TCR transgenic, and are capable of expanding up the TCR transgenic cells. This particular TCR transgenic originates from VAT Tregs, and the expanded T cells are able to improve insulin sensitivity (as shown for the vTreg53 TCR transgenic strain in the original paper by Liu et al., 2018).

The work is technically well performed, and no major experimental concerns are identified. However the title and Abstract rather over-sell the data available. "Discovery of surrogate agonists for visceral fat Treg cells" implies that the agonists are broadly capable of activating visceral fat Treg cells, while they are only expanding up the vTreg53 TCR transgenic cells. Even when injected into highly inbred mice with the same MHC, most (7/10) mice showed no response (3/10 mice showed an expansion of tetramer-reactive cells, one of which is shown in the manuscript, and no metabolic tests were performed). The selective expansion of vTregs was only in mice that were given the TCR transgenic cells to begin with, i.e. synthetic antigen expands up TCR transgenic cells.

---

## [Author Response]

Reviewer #2:[…] 1) Figure 4—figure supplement 1. The authors show some data from tetramer experiments. Even though only 3/10 mice responded it may be worth including graphs showing the number of tetramer positive cells from the different tissues. It is notable that the proportion of tetramer positive amongst Tregs is quite high in the VAT considering that no enrichment was performed, suggesting that at least in the representative mouse this worked rather well and may require more attention.

We have now included the absolute number of tetramer positive cells from the Spl/LN and VAT from the 10 mice immunized with adjuvant/Fat1562 in the new Figure 4—figure supplement 1C.

2) Figure 5B. I think a bit more effort can be put into the analysis of the differences between the groups. Using similar analysis to 5A were there any changes to the genes in the previously defined VAT UP and DOWN signature genes? What were the top differentially expressed genes and do any of them have known function in Tregs. How do the authors interpret the increased Pparg mRNA in the Spl/LN? Is this evidence of trafficking from the VAT? The authors note but do not show that other VAT signature genes may have increased.

We have now included volcano plots comparing gene expression between Tregs with and without peptide immunization in the spleen/LN and VAT (Figure 5D). We plotted top transcripts that either have known functions in Tregs, such as *Lag3, Il10, Tigit*, or those fall into the major pathways identified in GSEA, including *Isg15, Isg20* for interferon responses, *Nfil3 and Srebf2* for cholesterol metabolism, *E2f2, E2f7, E2f8*, and *Ccnb2* for E2F targets and G2M checkpoint. We have previously identified that the spleen hosts a small population of PPARγ^lo^ Tregs that are precursor cells for VAT Tregs (Li, et al., 2018), which is TCR-dependent. The RNA-seq data here fits with a model in which cognate antigen (TCR) stimulation upregulates *Pparg* expression and/or promotes the induction of VAT-Treg precursor cells in the spleen,. However, other interpretations are possible as well, e.g. a stimulus, like IL-4, that up-regulates *Pparg* expression. These different possibilities are under further investigation.

3) Figure 7G-M: Since the authors have already demonstrated that Fat1562 alone is effective in this system it seems hard to say if anti-TNF is making a significant addition to its efficacy since there is no Fat1562 only group for comparison. The normalized blood glucose level of the anti-TNF + Fat1562 group in Figure 7L is not noticeably better than the Fat1562 alone group in 7B. Similarly, comparison of 7I and 6G shows no support for a clear contribution of anti-TNF. Direct comparison within one experiment might be able to show a contribution of anti-TNFa to the effect of Fat1562 but this has not been done and without it I'm not sure this data is meaningful.

We had included only the 3 groups due to a dearth of mice, but agree with the reviewer that all four groups of mice (IgG, IgG+Fat1562, a-TNFα, a-TNFα+Fat1562) would be much better to dissect whether a-TNFα and Fat1562 immunization have a synergistic effect on expanding VAT Tregs and improving insulin sensitivity during obesity. We have now performed this experiment and include the results in the revised manuscript. In the new Figure 7H-K, we now show that in mice fed with long-term HFD, a-TNFα or Fat1562 alone had a modest effect on expanding VAT Tregs and improving insulin sensitivity, but the combination of a-TNFα and Fat1562 significantly increased the numbers of VAT Tregs, reduced the numbers of proinflammatory macrophages, and substantially improved insulin sensitivity.

4) It is also worth noting that similar data in Figures 6 and 7 are not presented consistently e.g. macrophage analysis is given as %CD11chi of F4/80 in 6H but #CD11chi/gram of fat in 7J. Raw numbers in 6E, x103 in 7I etc. These differences should be either justified or removed.

We have now plotted the numbers for eosinophils and inflammatory macrophages in the revised Figure 6H to keep it consistent with other figure panels.

Reviewer #3:In this manuscript the authors screen for peptides with the capacity to activate the vTreg53 TCR transgenic. The authors find several peptides that activate this TCR transgenic, and are capable of expanding up the TCR transgenic cells. This particular TCR transgenic originates from VAT Tregs, and the expanded T cells are able to improve insulin sensitivity (as shown for the vTreg53 TCR transgenic strain in the original paper by Liu et al., 2018).The work is technically well performed, and no major experimental concerns are identified. However the title and Abstract rather over-sell the data available. "Discovery of surrogate agonists for visceral fat Treg cells" implies that the agonists are broadly capable of activating visceral fat Treg cells, while they are only expanding up the vTreg53 TCR transgenic cells. Even when injected into highly inbred mice with the same MHC, most (7/10) mice showed no response (3/10 mice showed an expansion of tetramer-reactive cells, one of which is shown in the manuscript, and no metabolic tests were performed). The selective expansion of vTregs was only in mice that were given the TCR transgenic cells to begin with, i.e. synthetic antigen expands up TCR transgenic cells.

We would like to clarify that we have in fact observed expansion of the Fat1562/Ab tetramer+ Treg population in the spleen and lymph nodes of wild-type animals (c.f. Figure 4 and Figure 4—figure supplement 1B). As the reviewer notices, expansion of the Fat1562/A^b^ tetramer+ population was detected in 30% of wild-type mice. While this number may appear low, it is important to notice that TCRαβ sequences are generated randomly and are likely very different between mice. The 30% response rate in wild-type animals suggests that the v53Treg TCR clone is relatively common, which was unexpected. Furthermore, the single-cell clones that were isolated for TCR-sequencing showed that while one of the TCRs had a nearly identical CDR3α and CDR3β sequence to the vTeg53 TCR, other clones were different and lacked obvious similarities among them. Despite this, these TCRs were still able to recognize the Fat1562/A^b^ tetramer, suggesting that the VAT-Treg expansion is not necessarily limited to the vTreg53 clone.

As for the title, we understand that is it not perfect for reasons of length limitations, but readers will understand exactly what we have done from reading the Abstract and manuscript.